# Beyond Worst-Case Dimensionality Reduction for Sparse Vectors

**Sandeep Silwal**
UW-Madison
silwal@cs.wisc.edu

**David P. Woodruff**
CMU and Google
dwoodruf@andrew.cmu.edu

**Qiuyi Zhang**
Google DeepMind
qiuyiz@google.com

## Abstract

We study beyond worst-case dimensionality reduction for $s$-sparse vectors (vectors with at most $s$ non-zero coordinates). Our work first presents pessimistic lower bounds and then bypasses them by introducing a non-negative assumption:

(a) We first consider average-case guarantees for embedding $s$-sparse vectors. Here, a well-known folklore upper bound based on the birthday-paradox states: For any collection $X$ of $s$-sparse vectors in $\mathbb{R}^d$, there exists a linear map $A : \mathbb{R}^d \to \mathbb{R}^{O(s^2)}$ which *exactly* preserves the norm of 99% of the vectors in $X$ in any $\ell_p$ norm (as opposed to the usual setting where guarantees hold for all vectors). We provide novel lower bounds showing that this is indeed optimal in many settings. Specifically, any oblivious linear map satisfying similar average-case guarantees must map to $\Omega(s^2)$ dimensions. The same lower bound also holds for a wider class of sufficiently smooth maps, including 'encoder-decoder schemes', where we compare the norm of the original vector to that of a smooth function of the embedding. These lower bounds reveal a surprising separation result for smooth embeddings of sparse vectors, as an upper bound of $O(s \log(d))$ is possible if we instead use arbitrary functions, e.g., via compressed sensing algorithms.

(b) Given these lower bounds, we specialize to sparse *non-negative* vectors with hopes of improved upper bounds. For a dataset $X$ of non-negative $s$-sparse vectors and any $p \geq 1$, we show that we can non-linearly embed $X$ to $O(s \log(|X|s)/\varepsilon^2)$ dimensions while preserving all pairwise distances in $\ell_p$ norm up to $1 \pm \varepsilon$, with no dependence on $p$. Surprisingly, the non-negativity assumption enables much smaller embeddings than arbitrary sparse vectors, bypassing our previous quadratic lower bounds and improving on previous best known upper bounds that suffer an exponential $(\log |X|)^{O(p)}$ dependence. Our map also guarantees *exact* dimensionality reduction for the $\ell_\infty$ norm by embedding $X$ into $O(s \log |X|)$ dimensions, which is tight. We further give separation results showing that both the non-linearity of $f$ and the non-negativity of $X$ are necessary, and provide downstream algorithmic improvements using our embedding.

## 1 Introduction

Many popular algorithms for data processing in machine learning suffer from large running times on high-dimensional datasets. To alleviate this *curse of dimensionality*, a common paradigm is to first embed the data into a lower dimension and then run any desired algorithm in the embedded space. Arguably the most fundamental result in this area is the the Johnson-Lindenstrauss (JL) lemma. It states that any set of high-dimensional data $X$ with $|X| = n$ can be embedded into an $O(\log n/\varepsilon^2)$-dimensional space while approximately preserving all pairwise $\ell_2$ distances up to $\varepsilon$ relative error Johnson & Lindenstrauss (1984). While very versatile, the JL lemma is still a pessimistic worst case bound. This has led to a proliferation of works studying better trade-offs in more constrained settings, with the goal of exploiting intrinsic structure within the problem or in the dataset to obtain smaller embedding dimensions beyond JL. This 'fine-grained' approach has found success in many domains including nearest neighbor search Indyk & Naor (2007); Andoni et al. (2018), clustering

Boutsidis et al. (2010); Makarychev et al. (2019); Becchetti et al. (2019); Narayanan et al. (2021); Izzo et al. (2021); Jiang et al. (2024), and numerical linear algebra Woodruff et al. (2014); Cohen et al. (2015).

In this paper, we consider *data sparsity* as the structure to exploit to obtain better dimensionality reduction. Sparsity is an ubiquitous property of datasets and plays a crucial role in many tasks across machine learning, statistics, and signal-processing. Theoretically, sparsity assumptions allow for dimensionality reduction *upper bounds* for sparse vectors in general $\ell_p$ norms. In contrast, virtually all the aforementioned progress to improve beyond JL has been limited to the $\ell_2$ norm[1]. However, all known embedding[2] results for sparse vectors rapidly degrade as $p$ increases. For $\ell_p$ norms, it is known that any linear dimensionality reduction for sparse vectors must suffer an exponential dependence on $p$ on the embedding dimension Zhu et al. (2015). In particular for the important $\ell_\infty$ case, all known bounds become vacuous.

However, if we relax to *average-case* guarantees and only require 99% of the pairs of distances in a dataset of sparse vectors to be preserved, then a well known folklore dimensionality reduction map is known to give excellent dimensionality reduction. It is based on the birthday-paradox map:

**Definition 1.1** (Birthday Paradox Map). *Consider the linear map $f : \mathbb{R}^d \to \mathbb{R}^m$ where every coordinate in $\{1, \ldots, d\} = [d]$ is mapped uniformly at random to one of $m$ coordinates. The coordinates in $[d]$ that are mapped to the same 'bucket' in $[m]$ are summed.*

For any fixed $s$-sparse vector $x \in \mathbb{R}^d$, the birthday paradox implies that if $m \geq Cs^2$ for a large constant $C$, then with probability at least 99%, all the support elements of $x$ do not collide. Thus with probability at least 99%, $\|f(x)\|_p = \|x\|_p$ holds for any fixed sparse $x$ [3]. Since $f$ is linear, this also implies the same statement for pairwise distances. Interestingly, this map guarantees *exact* dimensionality reduction (for most pairs), the embedding dimension does not depend on $p$, and the guarantees hold for arbitrary sparse vectors. The natural question following from this discussion is if we can improve upon the $O(s^2)$ upper bound of the birthday paradox map, under the same average-case guarantees. This is the first question we address.

> *(Q1) Is the birthday paradox map optimal for 'average-case' dimensionality reduction of general sparse vectors?*

In our contributions detailed in Section 2, we prove novel lower bounds demonstrating that the birthday paradox map is optimal in many natural settings. This prompts us to ask if conditions *in addition to* sparsity enable improved dimensionality reduction upper bounds. Towards this end, we analyze the novel setting of sparse *non-negative* vectors, i.e., sparse vectors whose non-zero coordinates are positive, to go beyond existing lower bounds. This additional constraint is motivated in two ways. Practically in many applications, many sparse vector datasets are naturally non-negative, such as one-hot vectors in recommendation datasets or ReLU activations in deep learning He et al. (2024). Theoretically, there exists other fundamental problems demonstrating separations between non-negative sparse vectors and arbitrary sparse vectors Bringmann et al. (2021; 2022); Jin & Xu (2024). Note that one cannot assume non-negativity by simply shifting the dataset, as shifting can destroy sparsity. Thus the second question we ask is:

> *(Q2) Can we obtain improved dimensionality reduction for non-negative sparse vectors in general $\ell_p$ norms, such as $\ell_\infty$?*

## 2 OUR CONTRIBUTIONS

We detail our contributions to each question (Question 1 in Section 2.1 and Question 2 in Section 2.2). Table 1 summarizes our lower bound results, Table 2 summarizes our upper bound results, and Table 3 lists applications of our upper bounds. Additional related works are discussed in Section A.

---

[1]This is not for the lack of trying: there exist fundamental limits disallowing for dimensionality reduction for general $\ell_p$ norms Brinkman & Charikar (2005) for dense vectors.

[2]in this paper, we use the terms dimensionality reduction and embedding interchangeably.

[3]this hashing idea is ubiquitous, e.g. see CountMin and its variants Cormode & Muthukrishnan (2005)

## 2.1 Question 1: Average case lower bounds for general sparse vectors

Our main contribution towards (Q1) is to demonstrate that the average-case guarantees of the folklore birthday paradox upper bound of Definition 1.1 is *optimal* in many natural settings. First recall that the folklore upper bound guarantees that for any $p$ and for any point set of $s$-sparse vectors, there exists an embedding into $O(s^2)$ dimensions which preserves the $\ell_p$ norm *exactly* for 99% of the vectors (this is a weaker hypothesis than requiring pairwise distances to preserved as long as the all zeros vector is also mapped to the all zeros vector).

- **Arbitrary linear maps.** Our first result states that any linear map with arbitrary real entries which satisfies the properties of the folklore upper bound must embed to at least $m = \Omega(s^2)$ dimensions. Our lower bound holds for any even integer $p$. More precisely, we construct a set of $s$-sparse vectors in $d$-dimensions, where we show that any linear map $A$ which guarantees exact norm-preservation for more than 99% of this set (i.e., a randomly chosen vector in the set has a $> 99\%$ chance of their $\ell_p$ norm being preserved under the map) must map to $\Omega(s^2)$ dimensions. Note that the lower bound below for $d = s^2$ automatically implies the same lower bound for all higher $d$ by padding zeros in the 'hard' instance.

**Theorem 2.1** (Informal, see Theorem B.1). *Let $p \geq 2$ be an even integer. There exists a point set $S \subset \mathbb{R}^{s^2}$ of s-sparse vectors such that any linear map $A : \mathbb{R}^{s^2} \to \mathbb{R}^m$ satisfying $\|Ax\|_p = \|x\|_p$ for 99% of vectors $x \in S$ must map to $m = \Omega(s^2)$ dimensions.*

We note that our lower bound, even for the $p = 2$ case, is not implied by the known lower bounds for the JL lemma Alon (2003); Larsen & Nelson (2014; 2017), even if some of the hard point sets in prior works are sparse. This is because the hypotheses of the lower bounds are different. We prove a lower bound against only preserving 99% of distances, rather than all of them. Our lower bound also has no term that depends on the size of the point set (matching the folklore upper bound).

There is also a known lower bound for the JL lemma (again $p = 2$) where only one distance must be preserved with high probability (similar to the setting of the above theorem) Jayram & Woodruff (2013). However, this lower bound is only against *dense* vectors, and so it cannot imply Theorem 2.1. Prior lower bounds for the JL lemma are discussed in detail in Section A.

- **Beyond Exact Preservation.** Our lower bound for linear maps extends to a stronger statement for the $\ell_2$ case. We show that $m = \Omega(s^2)$ even if we only require the *weaker* guarantee of $\big| \|Ax\|_2^2 - \|x\|_2^2 \big| \leq O(\frac{\|x\|_2^2}{s})$. This is optimal as any larger relative error of $\varepsilon \gg 1/s$ can be accomplished with the standard linear JL map with $\tilde{O}(1/\varepsilon^2) = o(s^2)$ dimensions. This reveals an interesting phase transition: any larger error than $1/s$ enables $o(s^2)$ dimensional embeddings but any smaller error requires $\Omega(s^2)$ dimensional embeddings ($1/s$ error is no easier than 0 error!).

**Theorem 2.2** (Informal, see Theorem 3.1). *There exists a point set $S \subset \mathbb{R}^{s^2}$ of s-sparse vectors such that any linear map $A : \mathbb{R}^{s^2} \to \mathbb{R}^m$ satisfying $\big| \|Ax\|_2^2 - \|x\|_2^2 \big| \leq O(\|x\|_2^2/s)$ for 99% of vectors $x \in S$ must map to $m = \Omega(s^2)$ dimensions.*

We also extend this lower bound for linear maps to the case where we require inner-products to be preserved, rather than $\ell_2$ norms (Theorem B.4).

- **Arbitrary smooth embeddings.** For the $\ell_2$ case, we consider a more general class of embeddings $f : \mathbb{R}^{s^2} \to \mathbb{R}^m$ where $f(x) = (f_1(x), \dots, f_m(x))$ and each $f_i : \mathbb{R}^{s^2} \to \mathbb{R}$ is twice differentiable with continuous second partial derivatives. We prove that this large richer class of mappings still requires $m = \Omega(s^2)$ to satisfy the guarantees of the folklore upper bound for the $\ell_2$ case.

**Theorem 2.3** (Informal, see Theorem B.2). *The lower bound statement of Theorem 2.2 extends to the general class of mappings defined above.*

- **Beyond Embeddings.** Our $\ell_2$ lower bounds hold even when we are not restricted to computing the norm on the embeddings produced by a mapping $f$ and can use another function $g$ (on top of the output of $f$) to compute the norm. Formally, we define an encoding-decoding scheme using an *encoding* function $f$ which maps an $s$-sparse vector in $\mathbb{R}^{s^2}$ to a dimension $m$. A *decoding* function $g$ then takes the output of $f$ and maps it to a potentially much larger dimension $k$. Our goal is to

show that even if $k$ is large, as long as $m$ is substantially less than the initial sparsity squared, we cannot have $\|g(f(x))\|_2 \approx \|x\|_2$.

**Definition 2.1.** *We suppose the encoder and decoder functions satisfy the following.*

- *(Encoder function)* $f : \mathbb{R}^{s^2} \to \mathbb{R}^m$ *where*

$$f(x) = (f_1(x), \dots, f_m(x))$$

*and each* $f_i : \mathbb{R}^{s^2} \to \mathbb{R}$ *is twice differentiable with continuous second partial derivatives.*

- *(Decoder function)* $g : \mathbb{R}^m \to \mathbb{R}^k$ *where*

$$g(x) = (g_1(x), \dots, g_k(x))$$

*and each* $g_i : \mathbb{R}^m \to \mathbb{R}$ *is twice differentiable with continuous second partial derivatives.*

**Theorem 2.4** (Informal, see Theorem B.3). *The lower bound statement of Theorem 2.2 extends to the case of encoder/decoder schemes of Definition 2.1 and shows that $f$ (the encoder) must map to $\Omega(s^2)$ dimensions.*

Perhaps surprisingly, the theorem above states that if $f$ and $g$ are both sufficiently smooth, then $f$ still must map to $\Omega(s^2)$ dimensions (matching the folklore upper bound). That is, the whole procedure is still 'bottle-necked' by the dimension of $f$.

One reason for why this lower bound is surprising is that the hypothesis that $f$ and $g$ are smooth is quite crucial. In fact, the lower bound cannot hold if $g$ is arbitrary, even if $f$ is constrained to be a linear map. Indeed, compressed sensing algorithms tells us that there exists a suitable linear map $A : \mathbb{R}^d \to \mathbb{R}^{O(s \log(d))}$ (the so called RIP matrices Candes & Tao (2005)) and a decoding function $g$ such that $g(Ax) = x$ for any $s$-sparse $x$. However, $g$ is usually a complicated optimization step. For example in the popular Lasso algorithm for exact recovery, $g$ amounts to solving a linear program Foucart & Rauhut (2013), and the optimal solutions to such optimization programs can be highly discontinuous. If we also for arbitrary encoding functions $f$, then $f$ can also just be a list of the non-zero coordinates of $x$ along with their values, thereby also mapping into $\tilde{O}(s)$ dimensions. Clearly this $f$ is also highly discontinuous.

Thus, our results reveal an interesting separation result in the context of compressed sensing and sketching in general: if we let the encoding and decoding functions be arbitrary, then we can map to $\tilde{O}(s)$ dimensions. On the other hand, imposing even mild smoothness assumptions on $f$ and $g$ implies a much larger $\Omega(s^2)$ lower bound. See Section A for more details.

## 2.2 Question 2: Upper bounds for embedding non-negative sparse vectors

We begin with our upper bound theorem for embedding non-negative sparse-vectors.

**Theorem 2.5** (Informal, see Theorem C.2). *Let $X$ be a set of $n$ non-negative $s$-sparse vectors. For every $p \geq 1$, there exists a **non-linear** map $f : \mathbb{R}^d \to \mathbb{R}^m$ for $m = O(\log(ns) \cdot s/\varepsilon^2)$ which satisfies $\|f(x) - f(y)\|_p = (1 \pm \varepsilon)\|x - y\|_p$ for all $x, y \in X$. For the $\ell_\infty$ case, we can instead guarantee $\|f(x) - f(y)\|_\infty = \|x - y\|_\infty$ for all $x, y \in X$ with $m = O(s \log n)$.*

We note that for every $p$, it is the same map $f$ (up to scaling by a known factor) that gives the guarantees of the above theorem. There are three interesting aspects of Theorem 2.5 that we want to highlight:

1. The map $f$ is non-linear and non-smooth.

2. $X$ needs to be a point sets of non-negative sparse vectors.

3. Our embedding gives a non-trivial embedding for $\ell_\infty$.

Given these points, we additionally show that our upper bound for embedding non-negative sparse vectors presented in Theorem 2.5 is optimal in many ways.

| Point Set | Guarantee | Our Lower Bound | Thm. |
|---|---|---|---|
| $O(1)$-sparse non-negative vectors | $\frac{\|x\|_\infty}{2} \le \|f(x)\|_\infty \le \frac{3\|x\|_\infty}{2}$ 
 Linear $f$ | $\Omega(d)$ | 2.6 |
| $O(1)$-sparse non-negative vectors | $\|x+y\|_\infty \le \|f(x)+f(y)\|_\infty \le (2-\varepsilon) \cdot \|x+y\|_\infty$ 
 Arbitrary $f$, any $\varepsilon > 0$ | $\Omega(d)$ | 2.8 |
| $O(1)$-sparse non-negative vectors | $\forall r > 0, 0.99\|rx\|_\infty \le \|f(rx)\|_\infty \le 1.01\|rx\|_\infty$ 
 Arbitrary $f$ with continuous second-order derivatives | $\Omega(d)$ | 2.7 |
| $O(1)$-sparse vectors | $0.9 \cdot \|x-y\|_\infty \le \|f(x)-f(y)\|_\infty \le 1.1 \cdot \|x-y\|_\infty$ 
 $\|x+y\|_\infty \le \|f(x)+f(y)\|_\infty \le C \cdot \|x+y\|_\infty$ 
 Arbitrary $f$, any $C > 0$ | $\Omega(d)$ | 2.9 |
| $s$-sparse vectors | $\|f(x)-f(y)\|_\infty \le \|x-y\|_\infty$ 
 $C\|x-y\|_1 \le \|f(x)-f(y)\|_1$ 
 Arbitrary $f$, any $C > 0$ | $\Omega(Cs)$ | D.1 |
| $s$-sparse vectors | $\|f(x)\|_p = \|x\|_p$ for 99% of the points 
 Linear $f$, even $p$ | $\Omega(s^2)$ | B.1 |
| $s$-sparse vectors | $\|\|f(x)\|_2^2 - \|x\|_2^2\| \le O(\|x\|_2^2/s)$ for 99% of points 
 Arbitrary $f$ with continuous second-order derivatives | $\Omega(s^2)$ | B.2 |
| $s$-sparse vectors | $\|\|g(f(x))\|_2^2 - \|x\|_2^2\| \le O(\|x\|_2^2/s)$ for 99% of the points 
 Arbitrary $f$, $g$ with continuous second-order derivatives | $\Omega(s^2)$ | B.3 |

Table 1: Our lower bound results. $d$ is the ambient dimension. The first five guarantees hold for every pair of vectors $x, y$ in the point set specified in the respective theorems. The second to last column is the dimension that $f$ must map to. In the last row, $g$ is allowed to map to an arbitrary dimension.

• **Non-linearity is necessary.** We give an example of an $O(1)$-sparse and a non-negative point set in $\mathbb{R}^d$ where any *linear map* requires $\Omega(d)$ dimensions to preserve the $\ell_\infty$ norm up to relative error 0.5. This is particularly interesting as many prior *oblivious* dimensionality reduction maps, i.e. maps which do not depend on the input dataset such as JL or RIP matrices, are linear. However in the natural case of non-negative sparse vectors, linear maps are provably insufficient.

**Theorem 2.6.** *Let $S$ be the set of all $10$-sparse vectors in $\mathbb{R}^d$ with all non-zero coordinates equal to $1$. Let $A : \mathbb{R}^d \to \mathbb{R}^m$ be a matrix such that $\frac{1}{2}\|x\|_\infty \le \|Ax\|_\infty \le \frac{3}{2}\|x\|_\infty \ \forall x \in S$. Then $m = \Omega(d)$.*

• **'Non-smoothness' is necessary.** Our upper bound of Theorem 2.5 is continuous but not differentiable. Motivated by this phenomenon, we show that any upper bound satisfying Theorem 2.5 cannot be 'very smooth': any twice differentiable map (satisfying the same hypothesis) must embed into $\Omega(d)$ dimensions.

**Theorem 2.7.** *Suppose $f : \mathbb{R}^d \to \mathbb{R}^m$ is such that $f(x) = (f_1(x), \ldots, f_m(x))$ where each $f_i$ is twice differentiable with continuous second partial derivatives. Let $S$ be the set of all $10$-sparse vectors in $\mathbb{R}^d$ with all non-zero coordinates being equal to $1$. Suppose $f$ satisfies*

$$0.99\|rx\|_\infty \le \|f(rx)\|_\infty \le 1.01\|rx\|_\infty \quad \forall r > 0, \forall x \in S.$$

*Then $m = \Omega(d)$.*

• **Sums of non-negative sparse vectors cannot be preserved.** It is natural to ask if the map $f$ of Theorem 2.5 can also preserve *sums* of non-negative sparse vectors: $\|f(x) + f(y)\|_p \approx \|x + y\|_p$?

Indeed, our upper bound of Theorem 2.5 has the additional property that it preserves the norms of sums in $\ell_\infty$ norm up to a multiplicative factor of 2, even if we embed into *one* dimension (see the formal Theorem C.2). We show that the approximation factor 2 is tight in a very strong sense.

**Theorem 2.8.** *Let $e_i$ be the $i$th basis vector in $\mathbb{R}^d$. Consider the set $S = \{e_i\} \cup \{0\}$ of $d+1$ vectors. Suppose $f : S \to \mathbb{R}^m$ be an arbitrary mapping which satisfies*

$$\|x + y\|_\infty \leq \|f(x) + f(y)\|_\infty \leq (2 - \varepsilon)\|x + y\|_\infty \quad \forall x, y \in S$$

*for any $\varepsilon > 0$. Then $m \geq d$.*

• **General sparse vectors are hard to embed.** We also consider dropping the non-negative hypothesis. We again show that for the $\ell_\infty$ case, upper bounds as in the non-negative case cannot exist for general sparse vectors. Note that prior lower bounds studied in Zhu et al. (2015) are restricted to linear maps. In contrast, we show in the following theorem that any map which satisfies the same hypothesis as Theorem 2.5, but which holds for general sparse vectors with possibly *negative* entries, must embed into $\Omega(d)$ dimensions.

**Theorem 2.9.** *Let $e_i$ be the $i$th basis vector in $\mathbb{R}^d$. Consider the set $S = \{\pm e_i\} \cup \{0\}$ of $2d + 1$ vectors. Suppose $f : S \to \mathbb{R}^m$ be an arbitrary mapping which satisfies*

$$0.9\|x - y\|_\infty \leq \|f(x) - f(y)\|_\infty \leq 1.1\|x - y\|_\infty \quad \forall x, y \in S,$$

$$\|x + y\|_\infty \leq \|f(x) + f(y)\|_\infty \leq C\|x + y\|_\infty \quad \forall x, y \in S$$

*for any $C \geq 1$. Then $m \geq d$.*

The final lower bound implies that $\tilde{\Omega}(s)$ embedding dimension is necessary in Theorem C.2.

**Theorem 2.10** (Informal, see Theorem D.1). *Any map $f$ satisfying similar guarantees to that of Theorem C.2 with $\varepsilon = O(1)$ must map to $\Omega(s/\log(n))$ dimensions.*

| Approx. | $\ell_p$ norm | Our embedding dimension | Prior embedding dimension | Prior Reference |
|---|---|---|---|---|
| $1 \pm \varepsilon$ | $2 < p < \infty$ | $O\left(\frac{s \log(n)}{\varepsilon^2}\right)$ 
 Non-linear Map | $\frac{s^2}{\varepsilon^2} \cdot (p \log(n))^{O(p)}$ 
 Linear Map | Zhu et al. (2015) |
| | $1 < p < 2$ | | $O\left(\frac{s^p \log(d)}{\varepsilon^2} + \frac{s^{4-2/p-p} \log(d)}{\varepsilon^{2/(p-1)}}\right)$ 
 Linear Map | Zhu et al. (2015) |
| | $p \in \{1, 2\}$ | | $O\left(\frac{s \log(d/s)}{\varepsilon^2}\right)$ 
 Linear Map | Candes & Tao (2005); Berinde et al. (2008) |
| 1 | $p = \infty$ | $O(s \log(n))$ 
 Non-linear Map | - | - |

Table 2: Our upper bound results. The ambient dimension is $d$. Our upper bounds are in Theorem C.2, and hold for a set of $n$ non-negative $s$-sparse vectors. The first row is not explicitly given in Zhu et al. (2015), but can be derived from their argument. However, their result also holds for $n$ arbitrary $s$-sparse vectors. The $1 \leq p \leq 2$ cases of prior work can also handle all $s$-sparse vectors simultaneously.

## 2.3 APPLICATIONS OF OUR NON-NEGATIVE EMBEDDING

Our dimensionality reduction upper bound for non-negative sparse vectors, just as the JL lemma, has a slew of downstream algorithmic applications. We present a small subset of some of the new applications here, with a focus on more fundamental geometric optimization problems that have been well studied. At a high level, dimensionality reduction allows us to make black-box use of any

existing algorithm for a geometric task in low-dimensions. If the dimensionality reduction step is sufficiently powerful, we can hope to get faster runtimes at the cost of a small approximation factor loss; see Table 3 for a summary of our downstream algorithmic applications (diameter, max-cut, $k$-clustering, and more).

| Problem | Definition | Embedding Dimension | Distortion | Reference |
|---|---|---|---|---|
| Diameter | E.1 | $O(s^2)$ | $1$ | Lemma E.1 and Theorem E.3 |
| Max-Cut | E.2 | $O(s/\varepsilon^2)$ | $1 \pm \varepsilon$ | Theorem E.4 |
| $k$-median/$k$-center | E.3 | $O(s \log(n)/\varepsilon^2)$ | $4 \pm \varepsilon$ | Theorem E.5 |
| $k$-means | E.3 | $O(s \log(n)/\varepsilon^2)$ | $16 \pm \varepsilon$ | Theorem E.5 |
| Distance Estimation | E.4 | $O(s/\varepsilon^2)$ | $1 \pm \varepsilon$ | Theorem E.6 |

Table 3: Applications of our dimensionality reduction upper bound. The input is always a dataset of $n$ $s$-sparse non-negative vectors and the underlying norm is $\ell_p$ for an arbitrary $p \geq 1$. The distortion bounds hold with probability at least $99\%$. See the theorem statements for full details.

## 3 Technical Overview

We give on overview of our average-case lower bounds for embedding arbitrary sparse vectors and upper bound for embedding non-negative sparse vectors.

### 3.1 Overview: Average-Case Lower Bounds

In this section, we provide insights to our lower bound result of Theorem 3.1. Similar ideas are used in all of our average-case lower bounds, so we focus on this case which is the most instructive.

First we briefly outline how a known lower bounds for the JL lemma is proven. We state the proof of Alon (2003) where a slightly suboptimal lower bound is given (the result is tight up to $\log(1/\varepsilon)$ factors), but is perhaps more pedagogically useful in relation to our approach.

Alon (2003) construct a specific set of $n$ unit vectors and consider an embedding of these points into $m$ dimensions which preserves *all* pairwise distances up to $1 \pm \varepsilon$ multiplicative factor, or the inner product up to additive $\pm \varepsilon$. Letting $X \in \mathbb{R}^{n \times m}$ denote the matrix of embeddings, they consider the gram matrix $XX^T$. The key point is that they have precise control on *all* the entries of $XX^T$ due to the JL assumption. This allows them to argue that the rank of $X$ must be sufficiently large, implying a lower bound for $m$, the embedding dimension.

For us, we cannot directly employ this approach. Our lower bound hypothesis is weaker since we only guarantee that the norm is preserved say $99\%$ of the time. This means that we have no control over a constant fraction of the entries in $XX^T$, which can wildly influence the rank. However, this approach suggests that *rank* is a useful parameter and this observation is the starting point of our lower bound.

We now outline our lower bound approach for the case of a linear map $A$ given in Theorem 3.1. We first construct a suitable 'hard' input. We remark that showing a lower bound under this specific hard example extends to a lower bound for general point sets, as the former is a special case of the latter. The hard input we use can be represented as a distribution over sparse vectors. Our distribution first randomly samples the support elements and then puts random Gaussian values in the sampled support. The distribution we use, $\text{Unif}_{t,r}$, is defined below.

**Definition 3.1.** *To generate $x \sim \text{Unif}_{t,r}$,*

1. *First pick $t$ coordinates uniformly at random to be the support.*

2. *The non-zero coordinates of $x$ are mean zero i.i.d. Gaussians with variance $r$.*

If $r = 1$, we also denote the distribution as $\text{Unif}_t$. Theorem 3.1 states the following.

**Theorem 3.1.** *Let* $A : \mathbb{R}^{s^2} \to \mathbb{R}^m$ *be a linear map and* $\gamma \leq C/s$ *for a sufficiently small constant* $C > 0$. *If* $A$ *is such that for any* $1 \leq t \leq s$,

$$\Pr_{u \sim \text{Unif}_t} \left( |\|Au\|_2^2 - \|u\|_2^2| \leq \gamma \|u\|_2^2 \right) \geq 0.99,$$

*then* $m \geq s^2/1000$.

Note that as stated above, we are assuming the ambient dimension is $s^2$. This easily generalizes to any larger dimension by padding zeros.

To begin the proof, we first assume for contradiction that $m \ll s^2$. Now note that

$$\|Ax\|_2^2 - \|x\|_2^2 = \sum_{i,j} x_i x_j \langle A_i, A_j \rangle - \sum_i x_i^2 = \text{Tr}(xx^T A^T A) - \sum_i x_i^2$$

where $A_i$ are the columns of $A$. For simplicity, let's ignore the $\sum_i x_i^2$ portion for the rest of the discussion. We also assume the sum $\sum_{i,j} x_i x_j \langle A_i, A_j \rangle$ is over $i \neq j$. (In the formal proof, we show that the $\sum_i x_i^2$ portion and the $\sum_i x_i^2 \|A_i\|_2^2$ 'roughly cancel'.) We view this expression as a polynomial $P(x)$ of degree 2 in the variables $x_i$. Then the condition $|\|Ax\|_2^2 - \|x\|_2^2| \leq \gamma \|x\|_2^2$ implies that with large probability, $P(x)$ lies in a fixed interval $I$ of size $O(1)$ (as typically, $\gamma \|x\|_2^2 = O(1)$). Our goal now is to show that $P(x)$ has variance $\Omega(1)$ (under the randomness of $x$), implying it does not lie in any fixed $I$ with large constant probability, violating the hypothesis of the theorem.

Now it would be very bad for us if $P$ takes on 'very small' values over a typical choice of $x$. This means that we cannot rule out $P \in I$. For example, $P$ could even be equal to 0 if the *non-zero* entries of $x^T x$ only collide with the *zero* entries of $A^T A$. To avoid this possibility, we first demonstrate that a large fraction of the columns of $A$ have norms very close to 1. However, there are $s^2$ columns, which are all in $\mathbb{R}^m$. Since we assumed $m$ to be very small, this means that we have $\gg m$ almost-unit vectors in $m$ dimensions. We show that this implies there are at least $\Omega(s^2)$ pairs of $A_i$ and $A_j$ that have non-zero inner products (note that there are $\Theta(s^4)$ total pairs). This bound crucially relies on the fact that $m \ll s^2$ and can be thought of as a 'generalized' version of the rank argument used in Alon (2003). Since any pair $(i, j)$ is non-zero in $x^T x$ with probability approximately $1/s^2$, this means $x^T x$ and $A^T A$ 'collide' on a non-zero entry with a large constant probability.

However, this is not quite strong enough for our purposes since the coefficient of $P$, which are exactly $\langle A_i, A_j \rangle$, can be very close to 0 and unluckily, $x^T x$ could only collide on such small entries. Thus, our refined aim is to show that the coefficients of $P$ that collide with the non-zero entries of $x^T x$ are 'large' in the sense that the sum of these coefficient squared is $\Omega(1)$. It can be checked that this is sufficient to show the variance of $P(x) = \Omega(1)$.

Then in the formal proof, we show that the sum of non-zero coefficient squared is $\Omega(1)$, via a 'fine-grained' exploitation of the fact that $m \ll s^2$ (i.e. another appearance of rank). Now to argue anti-concentration of $P(x)$, we invoke the following classical inequality which states that the random variable $P(x)$ cannot be concentrated in an interval that is much smaller than the variance (which is directly related to the sum of non-zero coefficients squared).

**Lemma 3.2** (Theorem 8 in Carbery & Wright (2001)). *Let* $P : \mathbb{R}^n \to \mathbb{R}$ *be a degree* $d$ *polynomial and* $Z$ *denote the random variable where* $P$ *is evaluated on a standard Gaussian vector in* $\mathbb{R}^n$. *Then*

$$\Pr(|Z| \leq \varepsilon \sqrt{\text{Var}(Z)}) \leq O(d\varepsilon^{1/d}).$$

An application of Lemma 3.2 then implies that with sufficiently large constant probability, $P(x)$ *does not* lie in the interval $I$ defined above, contradicting the theorem hypothesis that $\|Ax\|_2^2$ approximately preserves the norm of $x$ with large probability. Hence, our assumption that $m \ll s^2$ is not valid, finishing the proof. The formal details are given in the proof of Theorem 3.1.

## 3.2 OVERVIEW: EMBEDDING NON-NEGATIVE SPARSE VECTORS

The goal of this section is to motivate our upper bound result of Theorem C.2. For simplicity, we only focus on the $\ell_\infty$ norm case of the theorem in the overview, but the ideas generalize to any $\ell_p$.

The main idea is to first construct a map with slightly weaker guarantees. We initially give a construction of a (randomized) map $f : \mathbb{R}^d \to \mathbb{R}^{O(s)}$ such that

1. $f$ preserves the $\ell_\infty$ norm of any pair in $X$ with probability 99%.

2. $f$ is never expanding deterministically: $\|f(x) - f(y)\|_\infty \leq \|x - y\|_\infty$ always.

Before discussing the construction of $f$, let's quickly see why the two points are beneficial towards the final construction. The final construction simply concatenates $O(\log n)$ independent copies of $f$. Due to independence, with high probability, every pair $x, y$ will have at least one copy of $f$ which 'certifies' the $\ell_\infty$ norm between them is at least $\|x - y\|_\infty$ (due to property (1)). Furthermore, since we know every copy of $f$ is non-expanding, we will never overestimate the distance (due to property (2)). Putting together these two statements implies the guarantees of our main theorem (for the $\ell_\infty$ case).

Now we describe the construction for $f$ which satisfies the two properties listed above. Property (1) actually holds for the birthday paradox map. However, this will not ultimately work since it cannot guarantee property (2). (And more directly, our lower bound in Theorem 2.6 rules out all linear maps). Instead, we use a *highly non-linear* map $f$. Similar to the birthday paradox map, we start by hashing all coordinates of the ambient dimension, $d$, to a set of $O(s)$ buckets. These buckets are the coordinates of the embedding. The crucial difference is that instead of summing the coordinates that land in a bucket, we take the *maximum*, a very unintuitive operation. More precisely, for a sparse vector $x$, we look at the buckets where the support elements of $x$ land. Then for all buckets which are non-empty, we take the maximum of the support elements of $x$ that land in the bucket, while all empty buckets get $0$.

The hashing step already guarantees property (1), similar to the birthday paradox upper bound. So it remains to check property (2). This reduces to checking the following: suppose coordinates $1, \ldots, k$ map to the same bucket under $f$. Let $x$ and $y$ be two non-negative sparse vectors. Then we want $|\max(x_1, \ldots, x_k) - \max(y_1, \ldots, y_k)| \leq \max_i |x_i - y_i|$, where $x_i$ and $y_i$ are the $i$th entries of $x$ and $y$ (they may be $0$). We can check that this is sufficient to imply property (2). To show this claim, imagine all the coordinates $x_i$ and $y_i$ together on the real line. They must all be to the right of the origin due to the non-negativity constraint. The right most coordinate, say $x_1$ without loss of generality, is always closer to the rightmost coordinate among the $y$'s, than it is to $y_1$. So the claim follows.

This construction crucially relies on the non-negativity of the vectors. An explicit example where our proposed map fails for arbitrary sparse vectors is as follows: consider two sparse vectors $x = [-1, 0, \ldots, 0]$ and $y = [0, 1, 0, \ldots, 0]$. Clearly, $\|x - y\|_\infty = 1$. Now suppose the first two coordinates hash to the same bucket under $f$. Then the first coordinate of $f(x)$ will have coordinate $-1$, since we take the maximum of all the support elements of $x$ that land in the first bucket. In this trivial case, the bucket is a singleton. Similarly, the first coordinate of the embedding of $y$ will be $1$, so $\|f(x) - f(y)\|_\infty = 2$. Thus, property (2) does not hold as the distance expands. One could try to massage the map $f$ to fix this particular issue, but our Theorem 2.9 rules out the existence of *any map* which preserves the distances between arbitrary sparse vectors in $\ell_\infty$ norm.

### 3.3 Open Problems

We note some interesting questions that follow naturally from our work.

1. What is the right $\varepsilon$ and $s$ dependency for sparse non-negative dimension reduction (Theorem C.2)? Can we improve upon the $\tilde{\Omega}(s)$ lower bound of Theorem D.1 to also include $\varepsilon$ factors? For example, for every $p \geq 1$, can we construct a set of $n$ non-negative $s$-sparse vectors such that any embedding $f$ to $m$ dimensions which preserves all pairwise $\ell_p$ distances up to distortion $1 \pm \varepsilon$ requires $m = \Omega(s \log(n)/\varepsilon^2)$? Interestingly, such a lower bound cannot be true for the $p = 2$ case since JL already gives us an upper bound of $O(\log n/\varepsilon^2)$ (regardless if the vectors are sparse or dense), so we must look at other $p$'s. On the other hand, for the $p = 2$ case, the JL lower bound of Alon (2003) implies a $\Omega(\log(n)/(\varepsilon^2 \log(1/\varepsilon)))$ lower bound for arbitrary mappings (for a $1 \pm \varepsilon$ approximation) since their hard point set consists of basis vectors (i.e. $1$-sparse). Thus, we know something about the $\varepsilon$ dependency for the $p = 2$ case. Can we extend this lower bound to other $\ell_p$ norms? Note that our 'average-case' lower bounds discussed in Section 3.1 are not directly applicable since they rule out exact (not approximate) dimensionality reduction for a constant fraction of (not all) pairs of vectors.

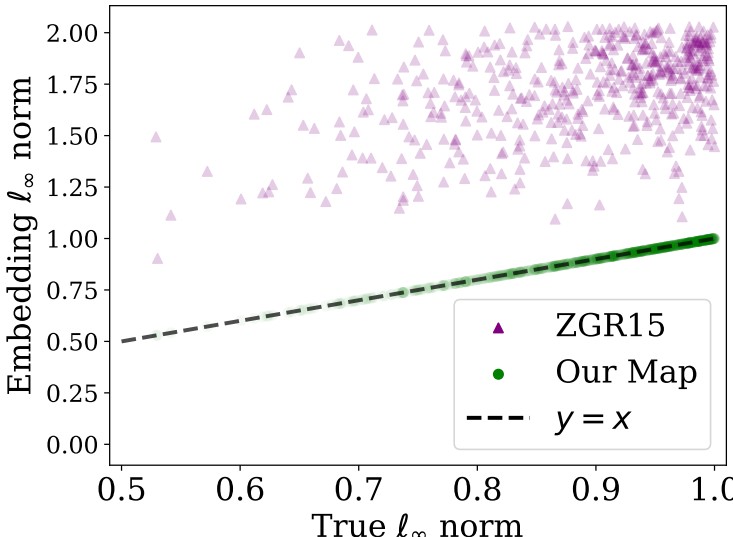

Figure 1: A simple plot demonstrating the performance of our non-negative embedding (Theorem C.2) versus the map of Zhu et al. (2015). The dots represent a 10-sparse vectors in $\mathbb{R}^{1000}$ with non-zero entries chosen uniformly in $[0, 1]$. The $x$-axis is the true $\ell_\infty$ norm and the $y$ axis is the approximated norm using the two different maps. Every vector has two dots (one for each map). We adapt both maps to embed to $\mathbb{R}^{50}$, but the performance is qualitatively similar for other $m$. We see that the performance of our map is demonstrably superior (it hugs the $y = x$ line).

2. Our 'average-case' lower bound of Theorem B.1 states that the folklore birthday-paradox mapping of Definition 1.1 is optimal for linear maps as well as sufficiently-smooth maps. It is an interesting question to prove a similar bound for arbitrary maps for the question of preserving 99% of distances between sparse vectors. Note that for linear maps, preserving distances is equivalent to preserving norms (on the set of differences of vectors), but this is not the case for arbitrary maps. The question of only preserving norms is easy for arbitrary maps: there exists an upper bound of 1 dimensions by just writing down the norm exactly, so the question of preserving distances is the natural one.

3. What is the power of *non*-linear embeddings in dimensionality reduction? Many *oblivious* dimensionality reduction upper bounds in literature use linear maps. Our results demonstrate a separation between linear and non-linear maps in the natural case of non-negative sparse vectors, and it is intriguing to ask if such separations exist in other settings.

4. In general, are there other natural dataset assumptions or relaxations that enable beyond worst case dimensionality reduction?

## 4 OUTLINE OF THE APPENDIX

Full details can be found in the appendix, which is organized as follows. Section A contains a discussion on related works. Section B gives full proofs for lower bounds for Question (1). Section C gives the full proofs for our upper bounds for Question (2). Section D proves that all of the hypothesis in our upper bound for embedding for non-negative sparse vectors are necessary. Lastly, Section E gives novel algorithmic applications based on our embedding for non-negative sparse vectors.

## 5 ACKNOWLEDGEMENTS

D. Woodruff worked on this in part at Google and while visiting the Simons Institute for the Theory of Computing. He also acknowledges support from a Simons Investigator Award.

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

# A    RELATED WORKS

**Lower-bounds for the JL Lemma**    Lower bounds for dimensionality reduction were introduced for understanding the minimum embedding dimension for $n$ vectors with at most $1 + \varepsilon$ multiplicative distortion. By using rank arguments of perturbed identity matrices, as outlined in Section 3.1, the first such lower bounds showed that the embedding dimension must satisfy $\Omega(\log(n)/(\varepsilon^2 \log(1/\varepsilon)))$, even when the embedded vectors are simple basis vectors Alon (2003). Furthermore, these bounds hold even when a non-linear or adaptive embedding function is applied; however they crucially depend on the *maximum* distortion being smaller than $\varepsilon$. These lower bounds were improved to an optimal $\Omega(\log(n)/\varepsilon^2)$ bound for an oblivious or fixed linear map Larsen & Nelson (2014), and then finally improved to any non-oblivious, non-linear embedding function by Larsen & Nelson (2017). It is worth noting that the final construction is significantly different that that of previous works and does not use nearly orthogonal sparse vectors. Note that these lower bounds inherently rely on the assumption that the dot product of the embedding must approximately preserve the dot product, without any post-processing or decoding.

**Speeding up JL for Sparse Vectors**    There have been a number of works on speeding up the runtime for embedding a collection of $n$ sparse vectors using a JL map (while still embedding to $O(\log(n)/\varepsilon^2)$ dimensions). Kane & Nelson (2014) demonstrates a distribution over sparse JL embedding matrices $\Pi$ such that $\Pi x$ takes $\tilde{O}(\|x\|_0/\varepsilon)$ time to compute, where $\|x\|_0$ denotes the number of non-zero entries of a vector $x$.

**Distributional Embedding Lower Bounds and JL**    Recall that the standard JL lemma states that for any $n$ vectors in $\mathbb{R}^d$, one can use a random Gaussian embedding to $O(\log(n)/\varepsilon^2)$ dimensions to guarantee the following maximum distortion bound with high probability: $(1 - \varepsilon)\|x - y\|^2 \leq \|Ax - Ay\|^2 \leq (1 + \varepsilon)\|x - y\|^2$ for all $x, y \in V$, where $|V| = n$. Furthermore, this embedding is also oblivious. While it is also known that the JL lemma is tight in the worst case, we emphasize that even our average-case lower bounds towards Question (2) (see Section 2.1) are not implied by the existing JL lower bounds, even for the $p = 2$ case of Theorem 2.1. At a high level, this is because our hypothesis is much weaker (we only require a large fraction of the norm to be preserved, rather than all pairs). We elaborate below.

JL embedding lower bounds state that for large enough $n$, there exists a *specific* point set on $n$ points such that *any map $f$* which preserves all pairwise distances must map to $\Omega(\log(n)/\varepsilon^2)$ dimensions Larsen & Nelson (2017). The main difference between this lower bound and our lower bounds of Section 2.1 are that the hypotheses assumed are different. The JL upper bound guarantee implies approximate norm preservation for *every* pair of differences of points in a collection of $n$ points, simultaneously and with high probability. Similarly, the JL lower bound assumes approximate norm preservation for *every* pair in a collection of $n$ vectors. On the other hand, the folklore birthday paradox upper bound assumes a fixed sparse vector as input, whose norm it preserves with constant probability. Similarly, our lower bounds of Section 2.1 only assume approximate norm preservation only a constant fraction of the time across a uniform distribution of suitably chosen sparse vector inputs. Consequently, we have no term depending on the number of input vectors in the statement (both in the folklore upper bound and the lower bounds).

There is also the *distributional* JL lemma which is a random map from $\mathbb{R}^d \to \mathbb{R}^{O(1/\varepsilon^2)}$ preserving the norm of any fixed vector in $\mathbb{R}^d$ up to a multiplicative $1 \pm \varepsilon$ with probability at least $99\%$ Johnson & Lindenstrauss (1984). It is shown in Jayram & Woodruff (2013) that the projection dimension of the distributional JL lemma is tight *information theoretically*. This ostensibly seems to imply our lower bounds (e.g. the $p = 2$ case of Theorem 2.1), for example if we parameterize the sparsity as $s = 1/\varepsilon$, seemingly rendering our lower bounds obsolete. But this is not the case! The information-theoretic lower bound proved in Jayram & Woodruff (2013) relies on *dense* vectors in $\mathbb{R}^d$. This is not an artifact of the proof, but an inherent requirement to prove their information-theoretic lower bound: any such lower bound *cannot* use sparse vectors. This is because information-theoretically, there is already a smaller projection dimension from compressed sensing: we can encode any sparse vector in $\tilde{O}(s)$ dimensions. Parameterizing the sparsity by $s = 1/\varepsilon$, this implies an information-theoretic upper bound of $\tilde{O}(1/\varepsilon)$, demonstrating that the lower bound of Jayram & Woodruff (2013) is not applicable to sparse vectors.

**Compressed Sensing** The field of compressed sensing studies decoding to recover encoded or compressed sparse vectors Donoho (2006). The restricted isometry property (RIP), introduced in Candes & Tao (2005), is a way to recover sparse vectors and focuses on bounding the RIP constant: the maximum distortion $D$ of a linear map $A \in \mathbb{R}^{m \times d}$ on all $s$-sparse vectors: ($\|x\|_p \leq \|Ax\|_p \leq D\|x\|_p$ for all $x$ with $x \in \mathbb{R}^d$ and $s$-sparse. The mappings under matrices with good RIP constants can serve as the compressed representation. Furthermore, matrices with such properties automatically give dimensionality reduction for $s$-sparse vectors. The prior known bounds for $m$ for RIP matrices with distortion $1 \pm \varepsilon$ are $O(s \log(d/s)/\varepsilon^2)$ for the $p = 1$ and 2 case Candes & Tao (2005); Berinde et al. (2008). For other values of $p$, Zhu et al. (2015) gave bounds of the form $\tilde{O}(s^p)$. If we only require to embed $n$ many $s$-sparse vectors, then one can calculate that Zhu et al. (2015)'s bound can be converted to $s^2(\log(n)p)^{O(p)}$, ignoring $\varepsilon$ factors (see Table 2). Thus, both bounds suffer an exponential dependence on $p$. Zhu et al. (2015) further showed that $m = \Omega(s^p)$ if $A$ has a bounded RIP constant.

We note that if the RIP constant is small enough, then applying a linear programming decoding via $\ell_1$ minimization recovers the sparse vectors when compressed to the informational-theoretical optimal $O(s \log(d))$ dimensions, with a matching lower bound Li et al. (2020); Mai et al. (2023). There are other alternatives, e.g. based on CountSketch Charikar et al. (2002); Price & Woodruff (2011), but all these decoders solve a complex optimization problem. Moreover, these mappings do not serve as embeddings/dimensionality reduction (e.g. the $\ell_p$ of the original vector is not approximated by the $\ell_p$ norm of the mapping).

**Role of Sparsity in Machine Learning** Primitives for sparse vectors, such as sparse matrix multiplication, is fundamental in a wide range of domains, such as graph analytics and scientific computing, and is used as iterative algorithms for sparse matrix factorization. Moreover, the field of deep learning often relies on faster sparse kernels to demonstrate speedups in practice, as it is a core operation in graph neural networks, transformers, and other architectures Child et al. (2019); Hoefler et al. (2021). On the theoretical side, there are also many works which seek to understand the role of sparsity in computational hardness for optimization problems such as sparse regression Natarajan (1995); Davis et al. (1997); Mahabadi (2015); Foster et al. (2015); Har-Peled et al. (2018); Chen et al. (2019); Gupte & Vaikuntanathan (2021); Price et al. (2022).

**Non-linear dimensionality reduction** One particularly interesting facet of our upper bound for embedding non-negative sparse vectors is that the mapping we give is non-linear. This is an example of an *oblivious* dimensionality reduction map (a mapping that does not depend on the input dataset) with this property. Oblivious maps include for instance JL matrices and RIP matrices, which are linear maps. We would like to highlight that non-linear dimensionality reduction results have found many applications, such as modifications of the JL lemma to $\ell_2$ distances raised to fractional powers Gottlieb & Krauthgamer (2015); Bartal et al. (2011), terminal embeddings for $\ell_2$ Cherapanamjeri & Nelson (2024); Narayanan & Nelson (2019); Mahabadi et al. (2018); Elkin et al. (2016), and finding the optimal way to embed a given set of vectors in euclidean space Grötschel et al. (2012). The last two of the aforementioned applications crucially adapt the mapping to a given input point set and are not oblivious.

# B   AVERAGE-CASE LOWER BOUNDS FOR EMBEDDING GENERAL SPARSE VECTORS

The goal of this section is to prove lower bounds showing that the birthday paradox map of Definition 1.1 is optimal in many settings. The point set we use are generated randomly from the following distribution as discussed in Section 3.

**Definition 3.1.** *To generate $x \sim \text{Unif}_{t,r}$,*

1. *First pick $t$ coordinates uniformly at random to be the support.*

2. *The non-zero coordinates of $x$ are mean zero i.i.d. Gaussians with variance $r$.*

Our first result is to show that any linear map with the same average-case guarantees as the birthday paradox map must map to $\Omega(s^2)$ dimensions. Note that the birthday paradox map satisfies the hypothesis of the theorem statement below up to constant factors (by mapping to $Cs^2$ dimensions for a sufficiently large constant $C$). This is because any fixed set of coordinates of size at most $s$ hashes to unique buckets under the map with probability say at least 99% (by picking large enough $C$). If this is the case, then it does not matter what entries we put in this set of coordinates.

**Theorem B.1.** *Let $A : \mathbb{R}^{s^2} \to \mathbb{R}^m$ be a linear map and $p \geq 2$ be an even integer. If $A$ is such that for any $1 \leq t \leq s$,*

$$\Pr_{u \sim \text{Unif}_t} (\|Au\|_p = \|u\|_p) \geq 0.99,$$

*then $m \geq s^2/1000$.*

*Proof.* Suppose for the sake of contradiction that $m < s^2/1000$. The proof is roughly divided into three parts. The first part shows that $A^T A$ has many 'non-zero' coordinates. The second part shows that a random sparse vector $u$ (as chosen in the hypothesis of the theorem statement) has 'many' pairs of coordinates $u_i$ and $u_j$ such that the corresponding $(i, j)$ entry in $A^T A$ is also non-zero. The last part then shows that the prior result implies that $A$ *does not* (approximately) preserve the norm of $u$ with a large constant probability, contradicting the assumption of the theorem. The last part relies on the fact that the zero set of a non-zero polynomial has measure 0.

**Part #1.** Let $A_1, \ldots, A_{s^2}$ denote the columns of $A$. Let $v$ be a random vector chosen from $\text{Unif}_1$, and let $i$ be its support with $v_i$ the corresponding non-zero entry. Then $\|Av\|_p^p = \|A_i\|_p^p \cdot v_i^p$ and $\|v\|_p^p = v_i^p$. Then the hypothesis implies that

$$\Pr(|\|A_i\|_2^2 - 1| \leq 0.00001) \geq 0.99. \tag{1}$$

(Note that we deliberately work with a weaker hypothesis since it will be more useful to us later on). Thus, a 0.99 fraction of the columns of $A$ have Euclidean norm in the range $[0.9999, 1.0001]$. Let $A_X$ be the restriction to such columns. Note that all diagonal entries of $A_X^T A_X$ are at in the range $[0.999, 1.001]$ and all entries are bounded by 1.001 in absolute value (via Cauchy-Schwarz on the columns of $A_X$). For $i \geq 1$, we let $\ell_i$ denote the number of *non-diagonal* entries of $A_X^T A_X$ whose absolute values are in $(2^{-i}, 2^{-i+1}]$ and $\ell_0$ denote the rest of the non-diagonal entries (with absolute values in $(1, 1.001]$). We show the following claims hold.

a) $\|A_X\|_F^2 \geq (0.99)^2 s^2$,

b) $\|A_X^T A_X\|_F^2 \geq \frac{(.99)^4 s^4}{m}$,

c) $\sum_{i \geq 0} \ell_i 2^{-2i+2} \geq \frac{(.99)^4 s^4}{m} - 1.001 s^2$.

Claim $(a)$ readily follows from inequality equation 1. To show $(b)$, note that singular values of $A_X^T A_X$ are the squared singular values of $A_X$ so $\|A_X^T A_X\|_F^2 = \sum_{i=1}^m \sigma_i(A_X)^4$ (since the rank of $A$ is at most $m$). By Cauchy–Schwarz,

$$m \cdot \|A_X^T A_X\|_F^2 = m \sum_i \sigma_i(A_X)^4 \geq \left( \sum_i \sigma_i(A_X)^2 \right)^2 = \|A_X\|_F^4 \geq (.99)^4 s^4.$$

To show (c), note that all diagonal entries of $A_X^T A_X$ are at in the range $[0.999, 1.001]$. Claim (c) then follows from using the lower bound of Claim (b) since there are at most $s^2$ diagonal entries.

**Part #2.** Now let $u$ be a vector drawn from $\mathrm{Unif}_s$ and let $T$ be its support. The hypothesis of the theorem states that

$$\Pr(\|Au\|_p = \|u\|_p) \geq 0.99. \tag{2}$$

Our goal is to demonstrate a contradiction by showing $\Pr(\|Au\|_p \neq \|u\|_p) \geq 0.02$, which contradicts equation 2.

Let $u_X$ be the restriction of $u$ to the coordinates in $X$ (i.e., only keep the coordinates in $X$). Consider $u_X u_X^T$. By our choice of $u$, the non-zero entries of $u_X u_X^T$ lie on a random principal submatrix of size $|T \cap X| \times |T \cap X|$. We show that with a sufficiently large constant probability, both $u_X u_X^T \in \mathbb{R}^{|X| \times |X|}$ and $A_X^T A_X \in \mathbb{R}^{|X| \times |X|}$ have a non-zero value in 'many' shared entries.

Towards this end, let $Y_{ij}$ be the indicator variable for the entry $(i, j)$ in $u_X u_X^T$ being non-zero and let $s_{ij}$ denote the *squared* $(i, j)$ entry of $A_X^T A_X$. We know

$$\mathbb{E}[Y_{ij}] = \frac{s(s-1)}{s^2(s^2-1)} = \frac{1}{s(s+1)}.$$

Finally, let $Z = \sum_{i,j} s_{ij} Y_{ij}$. Recalling our partitions $\ell_k$ that we defined earlier and Claim (c), we have

$$\mathbb{E}[Z] = \sum_{i,j} s_{ij}\, \mathbb{E}[Y_{ij}] \geq \frac{1}{s(s+1)} \sum_k \ell_k 2^{-2k} \geq \frac{1}{4s(s+1)} \left( \frac{(.99)^4 s^4}{m} - 1.001 s^2 \right) > 200.$$

We want to show $Z$ concentrates well around its mean. Towards this end, we bound $\mathbb{E}[Z^2]$. We have

$$Z^2 = \left( \sum_{i,j} s_{ij} Y_{ij} \right)^2 = \sum_{i,j} s_{ij}^2 Y_{ij} + \sum_{i,j,j'} s_{ij} s_{ij'} Y_{ij} Y_{ij'} + \sum_{i,j,i',j'} s_{ij} s_{i'j'} Y_{ij} Y_{i'j'}.$$

Since $s_{ij}^2 \leq 1.001$, the first term can be bounded as

$$\mathbb{E}\left[ \sum_{i,j} s_{ij}^2 Y_{ij} \right] \leq \frac{1.001}{s(s+1)} \sum_{i,j} s_{ij} = 1.001\, \mathbb{E}[Z].$$

Similarly, the third term can be bounded as

$$\mathbb{E}\left[ \sum_{i,j,i',j'} s_{ij} s_{i'j'} Y_{ij} Y_{i'j'} \right] = \sum_{i,j,i',j'} s_{ij} s_{i'j'} \left( \frac{1}{s(s+1)} \right)^2$$

$$\leq \left( \sum_{i,j} \frac{s_{ij}}{s(s+1)} \right)^2 = (\mathbb{E}[Z])^2.$$

It remains to bound the second term. We know $\mathbb{E}[Y_{ij} Y_{ij'}] \leq 1/s^3$. For a row $i$ of $A_X^T A_X$, let $Z_k(i)$ denote the number of entries in that row which are in level $\ell_k$. Then the second sum is

$$\mathbb{E}\left[ \sum_{i,j,j'} s_{ij} s_{ij'} Y_{ij} Y_{ij'} \right] \leq s \sum_i \sum_{j,j'} \frac{s_{ij} s_{ij'}}{s^4}$$

$$\leq 4s \sum_i \sum_{k,k'} \frac{Z_k(i) Z_{k'}(i)}{s^4 2^{2k} 2^{2k'}}$$

$$= 4s \sum_i \left( \sum_k \frac{Z_k(i)}{s^2 2^{2k}} \right)^2.$$

Let $t_i = \sum_k Z_k(i)2^{-2k}$. We now consider the following two cases.

**Case 1**: At least half of the $t_i$'s are at least $s$. In this case, we note that

$$\mathbb{E}[Z] \geq \sum_i \sum_k \frac{Z_k(i)}{2^{2k}s^2} = \sum_i \frac{t_i}{s^2} \geq \frac{|X|}{2s} \geq 0.495s.$$

We have

$$4s \sum_i \left( \sum_k \frac{Z_k(i)}{s^2 2^{2k}} \right)^2 \leq 4s \sum_i \sum_k \frac{Z_k(i)}{s^2 2^{2k}} \leq 4s\,\mathbb{E}[Z]$$

where we have used the fact that $\sum_k Z_k(i) \leq s^2$. So altogether,

$$\mathbb{E}[Z^2] \leq 5s\,\mathbb{E}[Z] + \mathbb{E}[Z]^2 \leq 12\,\mathbb{E}[Z]^2.$$

Thus the Paley–Zygmund inequality implies that

$$\Pr(Z \geq 1) \geq (1 - 0.01)^2 \cdot \frac{\mathbb{E}[Z]^2}{\mathbb{E}[Z^2]} \geq \frac{1}{20}.$$

**Case 2**: At least half of the $t_i$'s are at most $s$. In this case, let $A_{X'}^T A_{X'}$ be $A_X^T A_X$ restricted to the rows where $t_i \leq s$. We know $|X'| \geq 0.5|X|$. Our goal is to show that $A_{X'}^T A_{X'}$ still has at least 'many' non-zero off-diagonal entries.

The proof is identical to the proof of the three claims $(a), (b), (c)$ above. All sums below only pertain to the matrix $A_{X'}^T A_{X'}$. Indeed, we know $\|A_{X'}\|_F^2 \geq 0.495s^2$ so

$$m \cdot \|A_{X'}^T A_{X'}\|_F^2 = m \sum_i \sigma_i(A_{X'})^4 \geq \left( \sum_i \sigma_i(A_{X'})^2 \right)^2 = \|A_{X'}\|_F^4 \geq (.495)^2 s^4.$$

Combining with the fact that all non-diagonal entries are bounded by $1.001$ in absolute value, the above inequality implies that $\sum_{i \geq 0} \ell_i 2^{-2i+2} \geq (.495)^2 s^4/(1.001m) - s^2 \geq 50s^2$. Furthermore, for a row $i$ of $A_{X'}^T A_{X'}$,

$$\sum_k \frac{Z_k(i)}{s^2 2^{2k}} = \frac{t_i}{s^2} \leq \frac{1}{s},$$

so we can bound

$$4s \sum_i \left( \sum_k \frac{Z_k(i)}{s^2 2^{2k}} \right)^2 \leq 4 \sum_i \sum_k \frac{Z_k(i)}{s^2 2^{2k}}$$

$$= 4 \sum_i \frac{t_i}{s^2} \leq 4\,\mathbb{E}[Z].$$

Thus,

$$\mathbb{E}[Z^2] \leq 6\,\mathbb{E}[Z] + \mathbb{E}[Z]^2.$$

So by Payely-Zygmund,

$$\Pr(Z \geq 1) \geq (1 - 0.01)^2 \cdot \frac{\mathbb{E}[Z]^2}{\mathbb{E}[Z^2]} \geq \frac{1}{20}.$$

Thus we see that $Z \geq 1$ with probability at least $.05$ in both cases.

**Part #3.** Recalling that $T$ is the support set of $u$, we have $Au \sum_{t \in T} u_t A_t$ and thus,

$$\left\| \sum_{t \in T} u_t A_t \right\|_p^p - \|u\|_p^p = \sum_{i_1, \dots, i_p} u_{i_1} \cdots u_{i_p} \left( \sum_{j \in [m]} A_{i_1}(j) \cdots A_{i_p}(j) \right) - \sum_{t \in T} u_t^2 := P(u), \quad (3)$$

where the outer sum is over all $p$-sized tuples of indices (with repeats allowed), the notation $A_{i_1}(j)$ denotes the $j$th entry of the column $A_{i_1}$, and the inner sum is over all the entries of the column.

We claim that if $p$ is an even integer, then $P(u)$ is a non-zero polynomial. To do so, we demonstrate a monomial $u_{i_1} \cdots u_{i_p}$ with a non-zero coefficient. Now if we assume $Z \geq 1$, then this implies there exists two columns say $A_1$ and $A_2$ of $A$ such that $\langle A_1, A_2 \rangle$ is non-zero and both 1 and 2 are in the support set $T$ of $u$. We consider the following different cases for $p$.

If $p = 2$, consider the monomial $u_1 u_2$. It's coefficient in $P$ is $\sum_j A_1(j) A_2(j) = \langle A_1, A_2 \rangle \neq 0$, so we are done. Now if $p > 2$ and even, instead consider the monomial $u_1^{p-2} u_2^2$. We claim that this is non-zero. Indeed, since $p - 2$ is also even, $\sum_j A_1(j)^{p-2} A_2(j)^2$ must be non-zero since it is a sum of non-negative terms, one of which is non-zero (since at least one $j$ satisfies $A_1(j) A_2(j) \neq 0$). So we are also done.

Now finally, we note that since the entries of $u$ in its support are picked from a continuous distribution and $P$ is non-zero, the probability of the event $P(u) = 0$ is also $0^4$. However, $\|Au\|_p = \|u\|_p$ implies $\|Au\|_p^p - \|u\|_p^p = P(u) = 0$. Overall, with probability at least 0.05 (the event that $Z \geq 1$), we have $P(u) \neq 0$, contradicting the hypothesis equation 2. Thus, $m \geq s^2/1000$, as desired. $\qquad \square$

We extend the previous theorem to the case of *approximate* norm preservation for the $\ell_2$ case.

**Theorem 3.1.** *Let $A : \mathbb{R}^{s^2} \to \mathbb{R}^m$ be a linear map and $\gamma \leq C/s$ for a sufficiently small constant $C > 0$. If $A$ is such that for any $1 \leq t \leq s$,*

$$\Pr_{u \sim \text{Unif}_t} \left( |\|Au\|_2^2 - \|u\|_2^2| \leq \gamma \|u\|_2^2 \right) \geq 0.99,$$

*then $m \geq s^2/1000$.*

*Proof.* Suppose for the sake of contradiction that $m < s^2/1000$. The proof is roughly divided into three parts as in Theorem B.1. The first part shows that $A^T A$ has many 'non-zero' coordinates. The second part shows that a random sparse vector $u$ (as chosen in the hypothesis of the theorem statement) has 'many' pairs of coordinates $u_i$ and $u_j$ such that the corresponding $(i, j)$ entry in $A^T A$ is also non-zero. The last part then shows that the prior result implies that $A$ *does not* (approximately) preserve the norm of $u$ with a large constant probability, contradicting the assumption of the theorem. The last part relies on bounding the probability that a random polynomial lies in an unexpectedly small interval. Since the first two parts are identical, we just present the last part.

Recall the random variable $Z$ defined in the proof of Theorem B.1. There we showed that $Z \geq 1$ with probability at least 0.05.

**Part #3.** Recalling that $T$ is the support set of $u$, we have $\|Au\|_2^2 - \|u\|_2^2 = \left\|\sum_{t \in T} u_t A_t\right\|_2^2 - \|u\|_2^2$.

$$\left\|\sum_{t \in T} u_t A_t\right\|_2^2 - \|u\|_2^2 = \sum_{t \neq t' \in T} u_t u_{t'} \langle A_t, A_{t'} \rangle + \sum_{t \in T} \epsilon_t u_t^2 := P(u) \qquad (4)$$

where $|\epsilon_t| \leq 0.001$. If we pick the entries of $u$ to be standard Gaussians, we have

$$\mathbb{E}[P(u)] = \sum_t \epsilon_t.$$

Recalling that $Y_{ij}$ is the indicator for $u_X^T u_X$ having a non-zero entry at $(i, j)$, we have

$$\mathbb{E}[P(u)^2] \geq \sum_{i,j} s_{ij} Y_{ij} + \sum_{t \neq t' \in T} \epsilon_t \epsilon_{t'} + \sum_{t \in T} 3\epsilon_t^2$$

so

$$\text{Var}(P(u)) = \mathbb{E}[P(u)^2] - (\mathbb{E}[P(u)])^2 \geq \sum_{i,j} s_{ij} Y_{ij} = Z.$$

Thus, if we assume that $Z \geq 1$, then Lemma 3.2 implies that if the variables of $u$ in the support $T$ are picked from the standard Gaussian distribution, then the probability that $P(u) \in [-\varepsilon, \varepsilon] = O(\sqrt{\varepsilon})$ for any sufficiently small $\varepsilon > 0$. Now consider the following three events.

---

[4]e.g. see (https://math.stackexchange.com/users/822/nate eldredge)

- $\mathcal{E}_1$ = event that $\|u\|_2^2 \leq 100s$.

- $\mathcal{E}_2$ = event that $Z \geq 1$.

- $\mathcal{E}_3$ = event that $P(u) \notin [-c, c]$ for a sufficiently small constant $c$.

By picking $c$ to be a small enough constant, we know that *all* the events hold with probability at least 0.02. Let's condition on all of these events holding. If so, we show that the condition $\|\|Au\|_2^2 - \|u\|_2^2\| \leq \gamma\|u\|_2^2$ cannot hold. Indeed, the condition directly implies that $\|Au\|_2^2 - \|u\|_2^2 = P(u)$ lies in an interval strictly contained in $[-c, c]$. Thus altogether, with probability at least 0.02, we have $\|\|Au\|_2^2 - \|u\|_2^2\| > \gamma\|u\|_2^2$, contradicting inequality equation 2. This finishes the proof. $\quad\square$

We now extend the result of the prior theorem to a more general class of mappings. We recall the general class that we consider, as described in Section 2. We let $f : \mathbb{R}^{s^2} \to \mathbb{R}^m$ where $f(x) = (f_1(x), \ldots, f_m(x))$ and assume each $f_i : \mathbb{R}^{s^2} \to \mathbb{R}$ is twice differentiable with continuous second partial derivatives.

At a high level, the level of smoothness assumed allows us to consider a taylor expansion, where we approximate each $f_i$ using a linear function up to some quadratic error. By taking the expansion sufficiently close to the origin, the quadratic error becomes negligible, thereby reducing the problem to the linear case. Crucially, we use the fact that our sparse vectors are drawn from a 'scale invariant' distribution, in the sense that scaling a vector sampled from $\mathrm{Unif}_{t,r}$ is equivalent to sampling from $\mathrm{Unif}_{t,r'}$ for an appropriate $r'$. The full details are given in the proof below.

**Theorem B.2.** *Suppose $\gamma \leq C/s$ for a sufficiently small constant $C > 0$. If $f : \mathbb{R}^{s^2} \to \mathbb{R}^m$ as defined above is such that for any $1 \leq t \leq s$ and any $r > 0$,*

$$\Pr_{x \sim \mathrm{Unif}_{t,r}} \left(\|\|f(x)\|_2^2 - \|x\|_2^2\| \leq \gamma\|x\|_2^2\right) \geq 0.999,$$

*then $m \geq s^2/1000$.*

*Proof.* Note that we may assume that $f_i(0) = 0$; otherwise our guarantees would trivially fail when $r \to 0$. Since the second partial derivatives of all $f_i$ which comprise $f$ are continuous, they are bounded in magnitude in $[-1, 1]^{s^2}$, a compact set. Let $L$ be such an upper bound which holds for all $i$ (note that $L$ may depend on $s$). Now we let $c \ll 1$ be a sufficiently small value which will be determined shortly. For any fixed $f_i$, Taylor's theorem for multivariate functions[5] implies that for any $x \in [-c, c]^{s^2}$,

$$|f_i(x) - \langle \nabla f_i(0), x \rangle| \leq L(x_1 + \ldots + x_{s^2})^2 \leq Ls^4c^2$$

for every $i$. Let $A : \mathbb{R}^{s^2} \to \mathbb{R}^m$ be the matrix with $a_i$ as it's rows. The above inequality implies that

$$\|f(x) - Ax\|_\infty \leq c^2s^4L.$$

Thus Lemma F.3 implies that

$$\|\|f(x)\|_2^2 - \|Ax\|_2^2\| \leq c^3 \cdot \mathrm{poly}(s, L) \tag{5}$$

for all $x \in [-c, c]^{s^2}$ satisfying $\|f(x)\|_2^2 \leq 2\|x\|_2^2$. By picking $c$ sufficiently small, we can say the following:

- For any $x \in [-c, c]^{s^2}$ such that $\|\|f(x)\|_2^2 - \|x\|_2^2\| \leq \gamma\|x\|_2^2$, we also have $\|\|Ax\|_2^2 - \|x\|_2^2\| \leq \gamma\|x\|_2^2 + c^3 \cdot \mathrm{poly}(s, L)$,

- For any $x$ where $\|\|Ax\|_2^2 - \|x\|_2^2\| \leq \gamma'\|x\|_2^2$ holds for some scalar $\gamma' > 0$, then for any scalar $r > 0$, we also have $\|\|Ay\|_2^2 - \|y\|_2^2\| \leq \gamma'\|y\|_2^2$ where $y = r \cdot x$.

---

[5] `https://en.wikipedia.org/wiki/Taylor's_theorem#Taylor's_theorem_for_multivariate_functions`

The first claim follows from inequality equation 5, and the second claim follows from the fact that $A$ is a linear map. Now note that by picking a large enough constant $\beta \gg 1$, we know that sampling from any $t$, $x \sim \text{Unif}_{t,c/s^\beta}$ satisfies $\|x\|_2^2 \geq c^{2.5}/s^{\beta'}$ (for some other constant $\beta' > 0$) with probability $0.999$. Thus a sufficiently small choice of $c$ and large enough $\beta$ implies the following:

- For any $x \in [-c, c]^{s^2}$ such that $|\|f(x)\|_2^2 - \|x\|_2^2| \leq \gamma\|x\|_2^2$, we also have $|\|Ax\|_2^2 - \|x\|_2^2| \leq 2\gamma\|x\|_2^2$,

- For any $t$, $x$ sampled from $\text{Unif}_{t,c/s^\beta}$ is in $[-c, c]^{s^2}$ with probability at least $0.999$.

Now note that a uniformly chosen vector in $\text{Unif}_{t,r}$ for any $r > 0$ is just a scaled uniformly chosen vector in $\text{Unif}_{t,1}$. Due to our hypothesis on $f$, it then follows that $A$ is such that for any $1 \leq t \leq s$,

$$\Pr_{x \sim \text{Unif}_{t,1}} \left( |\|Ax\|_2^2 - \|x\|_2^2| \leq 2\gamma\|x\|_2^2 \right) \geq 0.99.$$

The theorem then follows from Theorem 3.1 (note that $2\gamma$ is *smaller* than the tolerance required in that proof). $\square$

As discussed in Section 2, we further extend our prior result to encoder decoder schemes (recalled below), where another 'decoder' function can be applied on top of the embeddings to compute the $\ell_2$ norm. Our lower bound shows that as long as both the encoder and decoder functions are sufficiently smooth, the encoder function is required to map to $\Omega(s^2)$ dimensions. We do not restrict the embedding dimension of the decoder function. In other words, the whole process is 'bottle necked' by the inner dimension. We recall the definition of encoder decoder schemes for convenience.

**Definition 2.1.** *We suppose the encoder and decoder functions satisfy the following.*

- *(Encoder function) $f : \mathbb{R}^{s^2} \to \mathbb{R}^m$ where*

$$f(x) = (f_1(x), \ldots, f_m(x))$$

  *and each $f_i : \mathbb{R}^{s^2} \to \mathbb{R}$ is twice differentiable with continuous second partial derivatives.*

- *(Decoder function) $g : \mathbb{R}^m \to \mathbb{R}^k$ where*

$$g(x) = (g_1(x), \ldots, g_k(x))$$

  *and each $g_i : \mathbb{R}^m \to \mathbb{R}$ is twice differentiable with continuous second partial derivatives.*

**Theorem B.3** (Encoder/Decoder Schemes). *Let $\gamma \leq C/s$ for a sufficiently small constant $C > 0$. If $h(x) = g(f(x)) : \mathbb{R}^{s^2} \to \mathbb{R}^{s^2}$ for $k = s^2$ (as defined above) is such that for any $1 \leq t \leq s$ and any $r > 0$,*

$$\Pr_{x \sim \text{Unif}_{t,r}} \left( |\|h(x)\|_2^2 - \|x\|_2^2| \leq \gamma\|x\|_2^2 \right) \geq 0.999,$$

*then $m \geq s^2/1000$.*

*Proof.* Similar to the proof of Theorem B.2, let $A$ be the matrix where the $i$th row is equal to $\nabla h_i(0)$. Note that $A : \mathbb{R}^{s^2} \to \mathbb{R}^{s^2}$. We first claim that $A$ has rank at most $m$. To see this, note that

$$h(x) = (g_1(f(x)), \ldots, g_{s^2}(f(x))).$$

For $1 \leq j \leq s^2$, let $h_j(x) = g_j(f(x))$. Denoting the input variables of $g_j$ as $g_j(y_1, \ldots, y_m)$, the chain rule tells us that for any $j$ and $i$,

$$\frac{\partial h_j}{\partial x_i} = \sum_{\ell=1}^m \frac{\partial g_j}{\partial y_\ell} \cdot \frac{\partial f_\ell}{\partial x_i} = \langle q_i, p_j \rangle$$

where

$$q_i = \left( \frac{\partial f_1}{\partial x_i}, \ldots, \frac{\partial f_m}{\partial x_i} \right) \in \mathbb{R}^m$$

and

$$p_j = \left( \frac{\partial g_j}{\partial y_1}, \dots, \frac{\partial g_j}{\partial y_m} \right) \in \mathbb{R}^m.$$

(For simplicity, we are omitting the fact that all the parital derivatives of $f_\ell$ are being evaluated at 0 and the parital derivatives of $g_j$ are being evaluated $f(0)$). Letting

$$B = \begin{bmatrix} - & q_1^T & - \\ - & q_2^T & - \\ & \vdots & \\ - & q_{s^2}^T & - \end{bmatrix} \in \mathbb{R}^{s^2 \times m}, \quad C = \begin{bmatrix} - & p_1^T & - \\ - & p_2^T & - \\ & \vdots & \\ - & p_{s^2}^T & - \end{bmatrix} \in \mathbb{R}^{s^2 \times m},$$

we see that

$$A = \begin{bmatrix} p_1^T B^T \\ p_2^T B^T \\ \vdots \\ p_{s^2}^T B^T \end{bmatrix} = C B^T \in \mathbb{R}^{s^2 \times s^2}.$$

Thus, $A$ has rank at most $m$. Now the rest of the proof proceeds by combining elements of Theorem 3.1 and B.2, which we only briefly sketch for simplicity.

First, identical to the proof of Theorem B.2, a second-order multi-variable talyor expansion around 0 implies that $A$ is such that for any $1 \le t \le s$

$$\Pr_{x \sim \text{Unif}_{t,1}} \left( \left| \|Ax\|_2^2 - \|x\|_2^2 \right| \le \gamma \|x\|_2^2 \right) \ge 0.99.$$

Now we cannot directly invoke Theorem 3.1, since the matrix $A$ in the statement of Theorem 3.1 maps $\mathbb{R}^{s^2}$ to $\mathbb{R}^m$, but $A$ is $\mathbb{R}^{s^2} \to \mathbb{R}^{s^2}$. However, note that the proof of Theorem 3.1 only relies on the *rank* of the matrix $A^T A$, which is at most $m$ (also true here). Thus, the rest of the proof is identical to the proof of Theorem 3.1. □

Finally, we extend the lower bound of Theorem 3.1 to the case of approximately preserving inner products.

**Theorem B.4.** *Let $A : \mathbb{R}^{s^2} \to \mathbb{R}^m$ be a non-zero linear map and $\gamma \le C/s$ for a sufficiently small constant $C > 0$. If $A$ is such that for any $1 \le t, t' \le s$,*

$$\Pr_{x \sim \text{Unif}_t, y \sim \text{Unif}_{t'}} \left( |\langle Ax, Ay \rangle - \langle x, y \rangle| \le \gamma \|x\|_2 \|y\|_2 \right) \ge 0.99$$

*and*

$$\Pr_{y \sim \text{Unif}_1} \left( \left| \|Ay\|_2^2 - \|y\|_2^2 \right| \le \gamma \|y\|_2^2 \right) \ge 0.99,$$

*then $m \ge s^2/1000$.*

*Proof.* The proof is almost identical to that of Theorem 3.1 but we present the full details for completeness, since we are working with two vectors instead of one. Suppose for the sake of contradiction that $m < s^2/10^3$. The proof is again roughly divided into three parts. The first part shows that $A^T A$ has many 'non-zero' coordinates. The second part shows that a pair of random sparse vector $u$ and $v$ (as chosen in the hypothesis of the theorem statement) have 'many' pairs of coordinates $u_i$ and $v_j$ such that the corresponding $(i, j)$ entry in $A^T A$ is also non-zero. The last part then shows that the prior result implies that $A$ *does not* (approximately) preserve the inner product with a large constant probability, contradicting the assumption of the theorem. The last part relies on bounding the probability that a random polynomial lies in an unusually small interval.

**Part #1.**

Let $A_1, \dots, A_{s^2}$ denote the columns of $A$. Let $v$ be a random vector chosen from $\text{Unif}_1$, and let $i$ be its support with $v_i$ the corresponding non-zero entry. Then $\|Av\|_2^2 = \|A_i\|_2^2 \cdot v_i^2$ and $\|v\|_2^2 = v_i^2$. Then the hypothesis implies that

$$\Pr(|\|A_i\|_2^2 - 1| \le 0.0001) \ge 0.99. \tag{6}$$

Thus, a 0.99 fraction of the columns of $A$ have Euclidean norm in the range $[0.9999, 1.0001]$. Let $A_X$ be the restriction to such columns. Note that all diagonal entries of $A_X^T A_X$ are at in the range $[0.999, 1.001]$ and all entries are bounded by $1.001$ in absolute value (via Cauchy-Schwarz on the columns of $A_X$). For $i \geq 1$, we let $\ell_i$ denote the number of *non-diagonal* entries of $A_X^T A_X$ whose absolute values are in $(2^{-i}, 2^{-i+1}]$ and $\ell_0$ denote the rest of the non-diagonal entries (with absolute values in $(1, 1.001]$). We show the following claims hold.

a) $\|A_X\|_F^2 \geq (0.99)^2 s^2,$

b) $\|A_X^T A_X\|_F^2 \geq \frac{(.99)^4 s^4}{m},$

c) $\sum_{i \geq 0} \ell_i 2^{-2i+2} \geq \frac{(.99)^4 s^4}{m} - 1.001 s^2.$

Claim $(a)$ readily follows from inequality equation 6. To show $(b)$, note that singular values of $A_X^T A_X$ are the squared singular values of $A_X$ so $\|A_X^T A_X\|_F^2 = \sum_{i=1}^m \sigma_i(A_X)^4$ (since the rank of $A$ is at most $m$). By Cauchy–Schwarz,

$$m \cdot \|A_X^T A_X\|_F^2 = m \sum_i \sigma_i(A_X)^4 \geq \left( \sum_i \sigma_i(A_X)^2 \right)^2 = \|A_X\|_F^4 \geq (.99)^4 s^4.$$

To show (c), note that all diagonal entries of $A_X^T A_X$ are at in the range $[0.999, 1.001]$. Claim (c) then follows from using the lower bound of Claim (b) since there are at most $s^2$ diagonal entries.

**Part #2.**

Now let $u$ and $v$ be a vector drawn from $\text{Unif}_s$ and $\text{Unif}_r$ respectively and let $T_u$ and $T_v$ be their corresponding support sets. The hypothesis of the theorem states that

$$\Pr(|\langle Au, Av \rangle - \langle u, v \rangle| \leq \gamma \|u\|_2 \|v\|_2) \geq 0.99. \tag{7}$$

Our goal is to demonstrate a contradiction by showing $\Pr(\langle Au, Av \rangle| > \gamma \|u\|_2 \|v\|_2) \geq 0.02$, which contradicts equation 7.

Let $u_X$ be the restriction of $u$ to the coordinates in $X$ (i.e., only keep the coordinates in $X$) and similarly define $v_X$. Consider $u_X v_X^T$. The non-zero entries of $u_X v_X^T$ lie on a random principal submatrix. We show that with a sufficiently large constant probability, both $u_X v_X^T \in \mathbb{R}^{|X| \times |X|}$ and $A_X^T A_X \in \mathbb{R}^{|X| \times |X|}$ have a non-zero value in 'many' shared entries.

Towards this end, let $Y_{ij}$ be the indicator variable for the entry $(i, j)$ in $u_X u_X^T$ being non-zero and let $s_{ij}$ denote the *squared* $(i, j)$ entry of $A_X^T A_X$. We know

$$\mathbb{E}[Y_{ij}] = \frac{1}{s^2}.$$

Finally, let $Z = \sum_{i,j} s_{ij} Y_{ij}$. Recalling our partitions $\ell_k$ that we defined earlier and Claim (c), we have

$$\mathbb{E}[Z] = \sum_{i,j} s_{ij} \mathbb{E}[Y_{ij}] \geq \frac{1}{s^2} \sum_k \ell_k 2^{-2k} \geq \frac{1}{4s^2} \left( \frac{(.99)^4 s^4}{m} - 1.001 s^2 \right) > 200.$$

The same proof as in Theorem 3.1 shows that $Z \geq 1$ with probability at least $.05$.

**Part #3.**

Recalling that $T_u$ and $T_v$ are the support sets of $u$ and $v$, we have

$$\langle Au, Av \rangle - \langle u, v \rangle = \sum_{i,j} u_i v_j \langle A_i, A_j \rangle - \sum_i u_i v_i := P(u, v). \tag{8}$$

If we pick the entries of $u$ and $v$ to be standard Gaussians, we have $\mathbb{E}[P] = 0$. Recalling that $Y_{ij}$ is the indicator for $u_X^T v_X$ having a non-zero entry at $(i, j)$, we have

$$\mathbb{E}[P^2] \geq \sum_{i,j} s_{ij} Y_{ij} = Z.$$

Thus, if we assume that $Z \geq 1$, then Lemma 3.2 implies that if the variables of $u$ and $v$ in their support are picked from the standard Gaussian distribution, then the probability that $P \in [-\varepsilon, \varepsilon] = O(\sqrt{\varepsilon})$ for any sufficiently small $\varepsilon > 0$. Now consider the following three events.

- $\mathcal{E}_1$ = event that $\|u\|_2 \|v\|_2 \leq 100s$.

- $\mathcal{E}_2$ = event that $Z \geq 1$.

- $\mathcal{E}_3$ = event that $P \notin [-c, c]$ for a sufficiently small constant $c$.

By picking $c$ to be a small enough constant, we know that *all* the events hold with probability at least 0.02. Let's condition on all of these events holding. If so, we show that the condition $|\langle Au, Av \rangle - \langle u, v \rangle| \leq \gamma \|u\|_2 \|v\|_2$ cannot hold. Indeed, the condition directly implies that $\langle Au, Av \rangle - \langle u, v \rangle = P$ lies in an interval strictly contained in $[-c, c]$. Thus altogether, with probability at least 0.02, we have $|\langle Au, Av \rangle - \langle u, v \rangle| > \gamma \|u\|_2 \|v\|_2$, contradicting inequality equation 7. This finishes the proof. □

## C  UPPER BOUNDS FOR EMBEDDING FOR NON-NEGATIVE SPARSE VECTORS

In this section we present our embeddings for non-negative sparse vectors and prove Theorem C.2. See section 3 for an overview.

Our embedding is constructed in two parts. In the first part, we introduce a 'base' mapping $f : \mathbb{R}^d \to \mathbb{R}^m$. It will be a random mapping which preserves *all* $\ell_p$ norms of a *fixed* pair of non-negative sparse vectors with constant probability. Crucially, it will be non-expanding. Our final embedding is formed by stacking many independent copies of our base mapping.

The base mapping maps every coordinate into one of $m$ buckets and takes the max of all non-zero coordinate elements that map to any bucket. The *max* pooling operation makes this map non-linear.

**Definition C.1** (Base mapping). *We define a mapping $f : \mathbb{R}^d \to \mathbb{R}^m$. Pick a uniformly random function from $h : [d] \to [m]$. For every $i \in [m]$, let $S_i = \{j \in [d] \mid h(j) = i\}$. We define $f(x) \in \mathbb{R}^m$ as follows. For every $i \in [m]$,*

$$f(x)_i = \begin{cases} \max(\{x_j \mid j \in S_i\}) & \text{if } S_i \neq \emptyset, \\ 0, & \text{otherwise.} \end{cases}$$

Our final embedding is the following. It stacks many copies of the base mapping. Note $\bigoplus$ denotes vector concatenation.

**Definition C.2** (Final Embedding). *A non-negative $(d, m, T)$ embedding $F : \mathbb{R}^d \to \mathbb{R}^{mT}$ is defined as follows. For every $1 \leq i \leq T$, let $f_i : \mathbb{R}^d \to \mathbb{R}^m$ be an independent copy of the random mapping of Definition C.1. Then for any $x \in \mathbb{R}^d$,*

$$F(x) = \bigoplus_{i=1}^{T} f_i(x).$$

The key properties of the base mapping are proved below. As discussed in Section 3, the non-expansion property is crucial and is only guaranteed for non-negative vectors due to the max operation.

**Theorem C.1.** *Let $f : \mathbb{R}^d \to \mathbb{R}^m$ be a random mapping of Definition C.1. It satisfies the following:*

1. *$f(0) = 0$ deterministically for all values $m \geq 1$.*

2. *For any pair of non-negative $s$-sparse vectors $x, y \in \mathbb{R}^d$ (both independent of $f$), if we take $m = \Omega(s^2/\delta)$,*
$$\Pr(\forall p, \|f(x) - f(y)\|_p = \|x - y\|_p) \geq 1 - \delta.$$

3. *For every pair of non-negative vectors $x$ and $y$ (not necessarily sparse) and any embedding dimension $m$, $\|f(x) - f(y)\|_p \leq \|x - y\|_p$ deterministically for all $p \geq 1$.*

*Proof.* The first property follow from the definition of $f$. Let $m = 100s^2/\delta$ and $h$ be the uniformly random function from $[d] \to [m]$ that constitutes $f$.

Let $x$ and $y$ be two fixed $s$-sparse vectors in $\mathbb{R}^d$. Proving the second condition relies solely on the fact that $h$ is likely to separate all the non-zero coordinates of $x$ and $y$. Indeed, the union of their supports is of size at most $2s$. Under $h$, the probability that some two domain elements in their union collide is at most

$$\binom{2s}{2} \cdot \frac{1}{(100s^2/\delta)} < \delta.$$

This event means that very coordinate in the union of the supports of $x$ and $y$ is mapped to a unique coordinate in $[m]$, and thus for every $p$,

$$\|x - y\|_p^p = \sum_{j=1}^d |x_i - y_i|^p = \sum_{i=1}^m |f(x)_i - f(y)_i|^p = \|f(x) - f(y)\|_p^p,$$

proving the second condition.

The third condition crucially relies on the fact that we are using the 'max' operation to define $f$. For any pair of non-negative vectors $x$ and $y$, we have

$$\|f(x) - f(y)\|^p = \sum_{i=1}^m |f(x)_j - f(y)_j|^p = \sum_{i=1}^m |\max(\{x_j \mid h(j) = i\}) - \max(\{y_j \mid h(j) = i\})|^p.$$

It suffices to prove the following: if $a_1, \ldots, a_k$ and $b_1, \ldots, b_k$ are non-negative real numbers then

$$|\max(a_1, \ldots, a_k) - \max(b_1, \ldots, b_k)| \leq \max_t |a_t - b_t|. \tag{9}$$

This is because assuming the claim, we have

$$\sum_{i=1}^m |\max(\{x_j \mid h(j) = i\}) - \max(\{y_j \mid h(j) = i\})|^p \leq \sum_{i=1}^m \max_{j|h(j)=i} |x_j - y_j|^p \leq \sum_{j=1}^d |x_i - y_i|^p,$$

since $h$ maps every coordinate in $[d]$ to only one coordinate in $[m]$.

Now to prove the claim, assume without loss of generality that $a_{t'} = \max(a_t) \geq \max_t(b_t)$. Then $|a_{t'} - \max_t(b_t)| \leq |a_{t'} - b_{t'}|$ since in the real line, we have the ordering $b_{t'} \leq \max_t(b_t) \leq a_{t'}$. $\square$

Building upon Theorem C.1, we give an embedding which approximately preserves all pairwise distances between points in a dataset of non-negative sparse vectors.

**Theorem C.2** (Embedding for non-negative sparse vectors). *Let $F : \mathbb{R}^d \to \mathbb{R}^{mT}$ be a non-negative $(d, m, T)$ embedding for $m = O(s/\varepsilon)$ and $T = O(\log(ns)/\varepsilon)$ as stated in Definition C.2. Let $X \subset \mathbb{R}^d$ be a dataset of $n$ non-negative $s$-sparse vectors (which is independent of $F$). We have*

1.
$$\Pr\left(\forall p \geq 1, \forall x, y \in X, \frac{\|F(x) - F(y)\|_p^p/T}{\|x - y\|_p^p} \in 1 \pm \varepsilon\right) \geq 1 - 1/\text{poly}(n).$$

2. *For the $\ell_\infty$ norm, it suffices to take $m = O(s)$ and $T = O(\log n)$ and guarantee that*
$$\Pr\left(\forall x, y \in X, \|F(x) - F(y)\|_\infty = \|x - y\|_\infty\right) \geq 1 - 1/\text{poly}(n).$$

3. *Additionally for the $\ell_\infty$ norm, it suffices to take $m = 1$ and $T = 1$ and guarantee that*
$$\Pr\left(\forall x, y \in X, \frac{\|F(x) + F(y)\|_\infty}{\|x + y\|_\infty} \in [1, 2]\right) = 1.$$

*Proof.* We first prove part (1). Note that

$$F = \bigoplus_{k=1}^{T} f_k$$

where each $f_k : \mathbb{R}^d \to \mathbb{R}^{O(s/\varepsilon)}$ is the mapping of Definition C.1 and $T = O(\log(ns)/\varepsilon)$. Now consider an arbitrary fixed pair $x, y \in X$. First by the non-expanding property of each $f_k$, we always have

$$\sum_k \|f_k(x) - f_k(y)\|_p^p \leq T \cdot \|x - y\|_p^p.$$

Now we show that the sum is also sufficiently large. Consider any fixed coordinate $i$ in the union of the support of $x$ and $y$. For any $f_k$, the probability that $i$ does not collide with any other support element is at least $1 - \varepsilon/100$ since there are only $O(s)$ total coordinates in the union of the supports of $x$ and $y$, and $f_k$ maps to $O(s/\varepsilon)$ dimensions. Since the $f_k$'s are independent, this means that $i$ will not collide with any other support elements for at least $1 - \varepsilon$ fraction of the $f_k$'s with failure probability at most $\exp(-\Omega(T\varepsilon))$ (Lemma F.1). By union bounding, we can extend this statement to *all* the coordinates in the support of $x$ and $y$, except with failure probability at most $s \exp(-\Omega(T\varepsilon)) \ll 1/\text{poly}(n)$, by our choice of $T$. Now if coordinate $i$ does not collide with any others in some $f_k$, then we know that $\|f_k(x) - f_k(y)\|_p^p \geq (x_i - y_i)^p$. We can sum this relation over all coordinates $i$ and all $f_k$. Thus, except with failure probability at most $1/\text{poly}(n)$, we have

$$\sum_k \|f_k(x) - f_k(y)\|_p^p \geq (1 - \varepsilon) T \cdot \|x - y\|_p^p,$$

as desired.

The $\ell_\infty$ case can be handled as follows. For a fixed pair $x, y$, suppose that the first coordinate witnesses their $\ell_\infty$ norm. If we take $m = O(s)$, then the first coordinate does not collide with any other coordinate with probability at least $99\%$ (there maybe collisions among other coordinates). Thus with large constant probability, a fixed base mapping $f$ of Definition C.1 certifies that $\|f(x) - f(y)\|_\infty \geq \|x - y\|_\infty$. But we always have $\|f(x) - f(y)\|_\infty \leq \|x - y\|_\infty$ deterministically. Thus with $O(\log n)$ repetitions, every pair $x, y$ will have at least one copy of the base mapping which ensures that the $\ell_\infty$ distance is *exactly* preserved.

For the third part of the theorem, note that since the coordinates are non-negative and $F(x) \in \mathbb{R}$ is just the maximum coordinate, we trivially have $\|F(x) + F(y)\|_\infty \geq \|x + y\|_\infty$. In the other direction, we claim that $\|f(x) + f(y)\|_\infty \leq 2\|x + y\|_\infty$ always holds deterministically where $f$ is our base mapping of Definition C.1. Indeed, it suffices to show that for any non-negative real numbers $a_1, \ldots, a_r$ and $b_1, \ldots, b_r$, we always have

$$\max(a_1, \ldots, a_r) + \max(b_1, \ldots, b_r) \leq 2 \max_{t \in [r]} (a_t + b_t). \tag{10}$$

This is seen to be true by just taking $t = \arg\max a_t$ (w.l.o.g. $\max(a_t) \geq \max(b_t)$). This completes the proof. $\qquad\square$

The *additive* guarantees of our mapping $F$ also extends to general $\ell_p$ norms, with an overhead of $2^{O(p)}$.

**Corollary C.3.** *Let $p \geq 1$ and $F : \mathbb{R}^d \to \mathbb{R}^{s^2 2^{O(p)} \log(n)/\varepsilon^2}$ be a non-negative $(d, s^2 2^{O(p)}/\varepsilon, T)$ embedding for $T = 2^{O(p)} \log(n)/\varepsilon$ as stated in Definition C.2. Let $X \subset \mathbb{R}^d$ be a dataset of $n$ non-negative $s$-sparse vectors (which is independent of $F$). We have*

$$\Pr\left( \forall x, y \in X, \frac{\|F(x) + F(y)\|_p^p / T}{\|x + y\|_p^p} \in 1 \pm \varepsilon \right) \geq 1 - 1/\text{poly}(n).$$

*Proof.* The proof is very similar to that of Theorem C.2 so we only highlight the differences. For the parameters, we take independent base embeddings $f_k : \mathbb{R}^d \to \mathbb{R}^m$ for $m = s^2 2^{O(p)}/\varepsilon$ (of Theorem C.1) and $T = \log(n) 2^{O(p)}/\varepsilon$. Let $\gamma = s^2/m$. First, for any fixed pair $x, y \in X$ we have

$\|f_k(x) + f_k(y)\|_p^p = \|x+y\|_p^p$ holds for at least $1 - \gamma$ fraction of $k$'s (due to property (2) in Theorem C.1) with failure probability at most $1/\mathrm{poly}(n)$. Thus we have

$$\sum_k \|f_k(x) + f_k(y)\|_p^p \geq (1-\gamma)T \cdot \|x+y\|_p^p.$$

To bound the other direction, we also always have $\|f_k(x) + f_k(y)\|_p^p \leq 2^p \|x+y\|_p^p$ for every $k$ from Eq. 10. This means that

$$\sum_k \|f_k(x) + f_k(y)\|_p^p \leq (1-\gamma)T \cdot \|x+y\|_p^p + \gamma T 2^p \|x+y\|_p^p.$$

Recalling the value of $\gamma$ finishes the proof as in Theorem C.2. □

If we limit the entries of the sparse vectors to be from a discrete set, then we can extend our non linear map to sparse vectors with arbitrary entries as well. Contrasting it with the known $\Omega(s^p)$ lower bounds for embedding arbitrary $s$-sparse vectors (using linear maps) from Zhu et al. (2015) hints that the 'hardness' for embedding arbitrary sparse vectors maybe due to entries with a large range.

**Theorem C.4.** *Let $p \geq 1$ and $F : \mathbb{R}^d \to \mathbb{R}^{s^2 \Delta^{O(p)} \log(n)/\varepsilon^2}$ be a $(d, s^2 \Delta^{O(p)}/\varepsilon), T)$ embedding for $T = \log(n)\Delta^{O(p)}/\varepsilon$ as stated in Definition C.2. Let $X \subset \mathbb{R}^d$ be a dataset of $n$ $s$-sparse vectors with entries in the discrete set $\{-\Delta, \ldots, \Delta\}$. We have*

$$\Pr\left(\forall x, y \in X, \frac{\|F(x) - F(y)\|_p^p}{\|x - y\|_p^p} \in [(1-\varepsilon)T, T]\right) \geq 1 - 1/\mathrm{poly}(n).$$

*Proof.* We again consider

$$F = \bigoplus_{k=1}^T f_k$$

where each $f_k : \mathbb{R}^d \to \mathbb{R}^m$ for $m = s^2(2\Delta)^p/\varepsilon$ is the mapping of Definition C.1 and $T = O(\log(n)(2\Delta)^p/\varepsilon)$. We can check that the first two properties of $f$ in Theorem C.1 hold for arbitrary $s$-sparse vectors. However, the third property crucially relies on non-negative entries. Nevertheless, we can obtain the following variant, assuming the entries are in a discrete set: for every pair of vectors $x, y \in \{-\Delta, \ldots, \Delta\}^d$, we have $\|f_k(x) - f_k(y)\|_p \leq \Delta \|x - y\|_p$. Indeed, as in the proof of Theorem C.1, it suffices to prove the following: if $a_1, \ldots, a_k$ and $b_1, \ldots, b_k$ are all in $\{-\Delta, \ldots, \Delta\}$, then

$$|\max(a_1, \ldots, a_k) - \max(b_1, \ldots, b_k)| \leq 2\Delta \cdot \max_t |a_t - b_t|.$$

And this holds because if the RHS is 0, then so is the LHS. Otherwise, the RHS is at least $2\Delta \cdot 1$ and the LHS is always bounded by $2\Delta$.

Equipped with this, we similarly have that with failure probability at most $\exp(-\Omega(T\gamma)) = 1/\mathrm{poly}(n)$, $\|f_k(x) - f_k(y)\|_p^p = \|x - y\|_p^p$ for at least $(1 - \gamma/2)T$ fraction of the indices $k$ where $\gamma = \varepsilon/(2\Delta)^p$. If this is the case, then again

$$\sum_k \|f_k(x) - f_k(y)\|_p^p \geq (1 - \gamma/2)T \cdot \|x - y\|_p^p$$

and by the bounded-expanding property of $f_k$, we also have

$$\sum_k \|f_k(x) - f_k(y)\|_p^p \leq (1 - \gamma/2)T \cdot \|x - y\|_p^p + \gamma T(2\Delta)^p \|x - y\|_p^p.$$

Putting everything together, we have $\sum_k \|f_k(x) - f_k(y)\|_p^p \in (1 \pm \varepsilon/2)T \cdot \|x - y\|_p^p$. And the final result follows from union bounding over all $\Theta(n^2)$ pairs, as desired. □

## D    LOWER BOUNDS FOR EMBEDDING NON-NEGATIVE SPARSE VECTORS

In this section, we provide lower bounds for dimensionality reduction for sparse vectors under various hypotheses. As stated in Section 2, together they demonstrate that our non-negative sparse embedding of Theorem C.2 is optimal in many natural ways. See Section 2 for an overview.

We begin by showing that a linear map cannot have the same guarantees as Theorem C.2.

**Theorem 2.6.** *Let $S$ be the set of all $10$-sparse vectors in $\mathbb{R}^d$ with all non-zero coordinates equal to $1$. Let $A : \mathbb{R}^d \to \mathbb{R}^m$ be a matrix such that $\frac{1}{2}\|x\|_\infty \leq \|Ax\|_\infty \leq \frac{3}{2}\|x\|_\infty \ \forall x \in S$. Then $m = \Omega(d)$.*

*Proof.* Suppose for the sake of contradiction that $m < d/100$. By considering the basis vectors, every column of $A$ must have an entry with absolute value at least $1/2$. Then there must exist a row $r$ of $A$ which has at least $d/m \geq 100$ such entries. At least $50$ of such entries in row $r$ must be of the same sign. Let $x$ be an indicator vector for the column of $10$ these entries in row $r$. Note that $x \in S$ and $|(Ax)_r| \geq 5$ which implies $\|Ax\|_\infty \geq 5$, contradicting our assumption on $A$. Thus, $m \geq d/100 = \Omega(d)$, as desired. $\qquad\square$

The following theorem states that preserving the norms of the sum of vectors is impossible a factor of $2 - \varepsilon$ for any $\varepsilon > 0$ in $\ell_\infty$, even if we only restrict to non-negative sparse vectors. Note that Theorem C.2 preserves the norms of the sums up to a factor of $2$.

**Theorem 2.8.** *Let $e_i$ be the $i$th basis vector in $\mathbb{R}^d$. Consider the set $S = \{e_i\} \cup \{0\}$ of $d+1$ vectors. Suppose $f : S \to \mathbb{R}^m$ be an arbitrary mapping which satisfies*

$$\|x + y\|_\infty \leq \|f(x) + f(y)\|_\infty \leq (2 - \varepsilon)\|x + y\|_\infty \quad \forall x, y \in S$$

*for any $\varepsilon > 0$. Then $m \geq d$.*

*Proof.* Suppose for the sake of contradiction that $m < d$. First note that $f(0)$ must be the all $0$'s vector by taking $x = y = 0$. Then taking $x = e_i$ and $y = 0$ implies that $\|f(e_i)\|_\infty \in [1, 2 - \varepsilon]$ for all $i$. Again label such coordinates of $f(e_i)$ that lie in this range as 'large.' If $m < d$, then there exists $i$ and $j$ such that $f(e_i)$ and $f(e_j)$ have the same large index by Pigeonhole. We know that $\|f(e_i) + f(e_j)\|_\infty \in [1, 2 - \varepsilon]$ by our hypothesis. Now if the large entries of $f(e_i)$ and $f(e_j)$ have the same sign, then their sum in absolute value is at least $2$, which cannot happen from the above observation. On the other hand, if they have different signs, then the largest sum (in absolute value) that these entries can add to is at most $1 - \varepsilon$ (either from $2 - \varepsilon + (-1)$ or $-(2 - \varepsilon) + 1$), which also cannot happen. These cases are exhaustive which means our assumption $m < d$ cannot hold, and we must have $m \geq d$. $\qquad\square$

The following theorem shows that any mapping $f$ also cannot be 'too smooth'. Note that in our mapping, we use the maximum function, which is not differentiable.

**Theorem 2.7.** *Suppose $f : \mathbb{R}^d \to \mathbb{R}^m$ is such that $f(x) = (f_1(x), \ldots, f_m(x))$ where each $f_i$ is twice differentiable with continuous second partial derivatives. Let $S$ be the set of all $10$-sparse vectors in $\mathbb{R}^d$ with all non-zero coordinates being equal to $1$. Suppose $f$ satisfies*

$$0.99\|rx\|_\infty \leq \|f(rx)\|_\infty \leq 1.01\|rx\|_\infty \quad \forall r > 0, \forall x \in S.$$

*Then $m = \Omega(d)$.*

*Proof.* Note that $f_i(0) = 0$ by taking $r \to 0$. Since the second partial derivatives of all $f_i$ are continuous, they are bounded in magnitude in the compact set $[0, 1]^d$. Let $T$ be such an upper bound which holds for all $i$. Now we let $r \ll 1$ be a sufficiently small value which will be determined shortly. For any fixed $f_i$, Taylor's theorem for multivariate functions[6] implies that for any $y \in [0, r]^d$ which is $O(1)$-sparse,

$$|f_i(y) - \langle \nabla f_i(0), y \rangle| \leq T (y_1 + \ldots + y_d)^2 \leq O(Tr^2)$$

---

[6] https://en.wikipedia.org/wiki/Taylor's_theorem#Taylor's_theorem_for_multivariate_functions

for every $i$. Let $A : \mathbb{R}^d \to \mathbb{R}^m$ be the matrix with $a_i$ as it's rows. The above inequality implies that
$$\|f(y) - Ay\|_\infty \le O(Tr^2).$$
Thus if $r > 0$ sufficiently small, $z$ is any vector such that $\|z\|_\infty = 1$, and
$$0.99\|rz\|_\infty \le \|f(rz)\|_\infty \le 1.01\|rz\|_\infty,$$
then we must also have
$$\frac{1}{2}\|rz\|_\infty \le \|A(rz)\|_\infty \le \frac{3}{2}\|rz\|_\infty.$$
But since $A$ is linear, this actually implies that
$$\frac{1}{2}\|r'z\|_\infty \le \|A(r'z)\|_\infty \le \frac{3}{2}\|r'z\|_\infty$$
for all $r' > 0$. Putting everything together, (letting $z$ be the vectors in the hypothesis of the theorem statement), implies that
$$\frac{1}{2}\|z\|_\infty \le \|Ax\|_\infty \le \frac{3}{2}\|z\|_\infty \quad \forall z \in S.$$
Then Theorem 2.6 implies that $m = \Omega(d)$, as desired. $\qquad\square$

The next lower bound of this section demonstrates that the non-negative hypothesis is crucial. If we drop the non-negativity constraint, then any map cannot satisfy the guarantees promised by Theorem C.2.

**Theorem 2.9.** *Let $e_i$ be the ith basis vector in $\mathbb{R}^d$. Consider the set $S = \{\pm e_i\} \cup \{0\}$ of $2d + 1$ vectors. Suppose $f : S \to \mathbb{R}^m$ be an arbitrary mapping which satisfies*
$$0.9\|x - y\|_\infty \le \|f(x) - f(y)\|_\infty \le 1.1\|x - y\|_\infty \quad \forall x, y \in S,$$
$$\|x + y\|_\infty \le \|f(x) + f(y)\|_\infty \le C\|x + y\|_\infty \quad \forall x, y \in S$$
*for any $C \ge 1$. Then $m \ge d$.*

*Proof.* Suppose for the sake of contradiction that $m \le d - 1$. We first show that $f(e_i) = -f(-e_i)$ for all $i$. Indeed, $e_i + (-e_i) = 0$ so we must have $\|f(e_i) + f(-e_i)\|_\infty = 0 \implies f(e_i) = -f(-e_i)$. We also have $f(0) = 0 \in \mathbb{R}^m$, also using the second relation. This means that $\|f(\pm e_i)\|_\infty \in [0.9, 1.1]$ for all $i$ by using the first relation with $x = \pm e_i$ and $y = 0$.

For a vector $v \in S$, call the indices where $\|f(v)\|_\infty \in [0.9, 1.1]$ *large entries*. Our goal is to show that most vectors in $S$ need to have distinct large entries. This will imply that $m$ must be large since $|S|$ is large.

Indeed, if $m \le d - 1$, then there must exists some $i \ne j$ such that $f(e_i)$ and $f(e_j)$ share the same large entry index by Pigeonhole. Then either $f(e_j)$ or $f(-e_j) = -f(e_j)$ also has the *opposite sign* as the large entry of $f(e_i)$. So at least in one case, we can find two vectors $x, y \in S$ such that $\|f(x) - f(y)\|_\infty \ge 2 \cdot 0.9 = 1.8$. However, since $i \ne j$, we know that $\|x - y\|_\infty = 1$, contradicting the first relation. Hence, $m \ge d$, as desired. $\qquad\square$

Finally, the last lower bound demonstrates that any map with weaker guarantees than that of Theorem C.2 must map to $\tilde{\Omega}(s)$ dimensions. Note that Theorem C.2 satisfies the hypothesis of the theorem bellow with $C = O(1/\log n)$ for a dataset of $n$ non-negative $s$-sparse vectors.

**Theorem D.1.** *Suppose $d > 2s$ and let $S = \left\{\sum_{i=1}^{s} e_i, \sum_{i=s+1}^{2s} e_i, 0\right\}$. Suppose $f : S \to \mathbb{R}^m$ be an arbitrary mapping which satisfies*
$$\|f(x) - f(y)\|_\infty \le \|x - y\|_\infty \quad \forall x, y \in S,$$
$$C\|x - y\|_1 \le \|f(x) - f(y)\|_1 \quad \forall x, y \in S,$$
*for any $C > 0$. Then $m \ge Cs$.*

*Proof.* By shifting, we can assume $f(0) = 0$. Let $v_1 = \sum_{i=1}^{s} e_i$ and $v_2 = \sum_{i=s+1}^{2s} e_i$. Then the first relation implies that all coordinates of $v_1$ and $v_2$ are bounded by 1 in absolute value. The second relation them implies that
$$2Cs \le \|f(v_1) - f(v_2)\|_1 \le \|f(v_1)\|_1 + \|f(v_2)\|_1 \le 2m,$$
yielding $m \ge Cs$. $\qquad\square$

# E  APPLICATIONS OF OUR NON-NEGATIVE EMBEDDING

We now present applications of our embeddings for sparse vectors for many problems in geometric optimization. We always assume our input is a dataset $X$ of $n$ non-negative $s$-sparse vectors in $\mathbb{R}^d$.

## E.1  COMPUTING DIAMETER

Our goal is to compute the diameter of the dataset $X$ in $\ell_p$ norm (see Definition E.1). The first lemma shows that the diameter is preserved when projecting onto very low dimensions.

**Definition E.1.** $\text{diameter}_p = \max_{x,y \in X} \|x - y\|_p$.

**Lemma E.1.** *Let $f : \mathbb{R}^d \to \mathbb{R}^{O(s^2)}$ mapping given in Definition C.1. Let $\tilde{X} = \{f(x) \mid x \in X\} \subset \mathbb{R}^{O(s^2)}$. We have*

$$\Pr\left(\forall p, \text{diameter}_p(X) = \text{diameter}_p(\tilde{X})\right) \geq 0.99.$$

*For the $\ell_\infty$ case, we can instead embed to $O(s)$ dimensions.*

*Proof.* Let $(x, y)$ be a pair in $X$ that witnesses the diameter. Theorem C.1 implies that for any $p$, the distance between $x$ and $y$ is preserved under $f$ with probability at least 99%. The proof of the $\ell_\infty$ case is the same as in Theorem C.2: the coordinate which witnesses the distance between $x$ and $y$ does not collide with any of the other support elements with constant probability. Furthermore, Theorem C.1 also implies that all other distances do not expand under $f$. This completes the proof. $\square$

This implies the following corollary stating that any 'low dimensional' algorithm for computing the diameter can be used to compute the diameter of $X$, after composing with our dimensionality reduction.

**Corollary E.2.** *Given $p \geq 1$, consider an algorithm which computes a $C$-factor approximation of the diameter of any $n$ point dataset in $d$-dimensions in $Q(n, d, C)$ time. Then there exists an algorithm which computes a $C$ approximation of the diameter correctly with 99% probability of our dataset $X$ in time $Q(n, O(s^2), C) + O(ns)$. For the $\ell_\infty$ case, the corresponding bound is $Q(n, O(s), C) + O(ns)$.*

*Proof.* We simply project our dataset $X$ using $f$ as stated in Lemma E.1 and apply the algorithm $Q$ in the projected space. $\square$

Appealing to existing algorithms on diameter computation implies the following results.

**Theorem E.3.** *Let $X$ be dataset of non-negative $s$-sparse vectors. We have the following algorithms:*

1. *Using Chan (2018), for any constant integer $p$, we can compute a $1 + \varepsilon$ approximation to $\text{diameter}_p$ of $X$ in time $\tilde{O}(n/\sqrt{\varepsilon} + ns + 2^{O(s^2 \log(1/\varepsilon))})$ which is correct with probability 99%.*

2. *We can exactly compute the diameter of $X$ in $\ell_\infty$ norm in time $O(ns)$. This algorithm can be implemented in a stream using $O(s)$ words of memory.*

3. *We can exactly compute the diameter of $X$ in $\ell_1$ norm in time $n2^{O(s)}$. This algorithm can be implemented in a stream using $2^{O(s)}$ words of memory.*

*Proof.* The first result just follows from appealing to Corollary 4.1 in Chan (2018) which gives an algorithm for computing the diameter in $\ell_p$ in low-dimensional spaces, and appropriately plugging in the bounds as specified in Corollary E.2.

For the second result, we recall a folklore streaming algorithm for computing diameter in $\ell_\infty$ norm. We have

$$\text{diameter}_p(X) = \max_{\text{dimension } i} \max_{x,y \in X} |x_i - y_i| = \max_{\text{dimension } i} \left(\max_{x \in X} x_i - \min_{y \in X} y_i\right).$$

Thus, it just suffices to only keep track the maximum and the minimum coordinate along every dimension. Via Lemma E.1, it suffices to assume the dimension is only $O(s)$. The non-streaming algorithm simply scans the points along every dimension as well.

Finally for the third result, we recall a well known isometric embedding of $\ell_1^k$ into $\ell_\infty^{2^k}$ for any $k \geq 1$: simply map any $x \in \mathbb{R}^d$ to $Ax \in \mathbb{R}^{2^k}$ where $A$ has all possible $\pm 1$ vectors as rows. Applying this embedding reduces the $\ell_1$ case to the $\ell_\infty$ case addressed above. $\qquad\square$

### E.2 DIMENSIONALITY REDUCTION FOR MAX CUT

We consider the max cut problem defined as follows.

**Definition E.2.** $\mathrm{MaxCut}_p(X) = \max_{S \subseteq X} \sum_{x \in S, y \in X \setminus S} \|x - y\|_p^p$.

To the best of our knowledge, dimensionality reduction for max cut has been only studied in the $\ell_2$ case where its known that $O(1/\varepsilon^2)$ dimensions suffice to estimate the max cut of an arbitrary sized dataset up to $1 \pm \varepsilon$ (only for $\ell_2$) Lammersen et al. (2009); Chen et al. (2023).

We show the following dimensionality reduction for max cut in the general $\ell_p$ case, where the point set consists of non-negative sparse vectors.

**Theorem E.4.** *Let $f : \mathbb{R}^d \to \mathbb{R}^{O(s/\varepsilon^2)}$ be the mapping given in Theorem C.1. Let $\tilde{X} = \{f(x) \mid x \in X\}$. For every $p \geq 1$, we have*

$$\mathbb{E}\left[\left|\mathrm{MaxCut}_p(\tilde{X}) - \mathrm{MaxCut}_p(X)\right|\right] \leq O(\varepsilon) \cdot \mathrm{MaxCut}_p(X).$$

*Proof.* We drop the dependence on $p$ for the sake of clarity. Let $E$ be the set of edges that participate in a fixed optimal max cut for $X$. Let $\overline{\mathrm{MaxCut}}(\tilde{X})$ be the value of the cut given by $E$ in the projected dimension (note that $E$ is deterministic but the value is random since the projection is random).

Fix any pair $x, y$ and consider the random variable $\|f(x) - f(y)\|_p^p$. We know deterministically $\|f(x) - f_1(y)\|_p^p \leq \|x - y\|_p^p$. For a non-zero coordinate $z_i$ of $z := x - y$, let $t_i$ denote the indicator random variable for the event that $i$ does not collide with any of the other non-zero coordinates of $z$. By our choice of $m$, we know that $\mathbb{E}[t_i] \geq 1 - \varepsilon^2$ since the sparsity of $z$ is $O(s)$, as it is the difference of two $s$-sparse vectors. Thus,

$$\mathbb{E}[\|f(x) - f(y)\|_p^p] \geq (1 - \varepsilon^2)\|x - y\|_p^p,$$

so by Markov's inequality,

$$0 \leq \|x - y\|_p^p - \|f(x) - f(y)\|_p^p \leq \varepsilon \|x - y\|_p^p$$

with probability at least $1 - \varepsilon$. We have

$$\mathbb{E}\left[\left|\mathrm{MaxCut}(X) - \overline{\mathrm{MaxCut}}(\tilde{X})\right|\right] \leq \sum_{(x,y) \in E} \mathbb{E}\left[|\|f(x) - f(y)\|_p^p - \|x - y\|_p^p|\right]$$

$$\leq \sum_{(x,y) \in E} O(\varepsilon) \cdot \|x - y\|_p^p$$

$$= O(\varepsilon) \cdot \mathrm{MaxCut}(X),$$

Now we make two simple observations. First, we always have

$$\mathrm{MaxCut}(\tilde{X}) \geq \overline{\mathrm{MaxCut}}(\tilde{X}),$$

since $\mathrm{MaxCut}(\tilde{X})$ is the best cut in the projected space. Secondly, since the value of every cut never expands in the projected dimension due to property (3) in Theorem C.1, we also have that the value of *every* cut in the projected space is at most $\mathrm{MaxCut}(X)$ deterministically. In particular, it must also be true that

$$\mathrm{MaxCut}(X) \geq \mathrm{MaxCut}(\tilde{X}).$$

In particular, this means that we have

$$0 \leq \mathrm{MaxCut}(X) - \mathrm{MaxCut}(\tilde{X}) \leq \mathrm{MaxCut}(X) - \overline{\mathrm{MaxCut}}(\tilde{X}),$$

so the random variable $\left|\mathrm{MaxCut}(\tilde{X}) - \mathrm{MaxCut}(X)\right|$ is always bounded by $\left|\mathrm{MaxCut}(X) - \overline{\mathrm{MaxCut}}(\tilde{X})\right|$, which finishes the proof. $\qquad\square$

### E.3 CLUSTERING APPLICATIONS

We consider the (arguably) most well-studied formulations of clustering: $k$-median, $k$-means, and $k$-center. Dimensionality reduction for these problems have been well studied in the $\ell_2$ case Makarychev et al. (2019). Here we consider dimensionality reduction for general $\ell_p$ norms, but with a restricted set of vectors (non-negative sparse).

We recall their definitions below.

**Definition E.3.** *We are interested in general $\ell_p$ norm formulations of the $k$-median/center/means clustering objectives.*

- *$k$-**median**: Given a dataset $X = \{x_1, \ldots, x_n\}$ of $n$ points $\in \mathbb{R}^d$, the goal is to find a partition $\mathcal{C} = \{C_1, C_2, \ldots, C_k\}$ of $[n]$ into $k$ non-empty parts (clusters) to minimize the following:*

$$\text{cost}(\mathcal{C}(X)) = \sum_{i=1}^{k} \min_{u_i \in \mathbb{R}^d} \sum_{j \in C_i} \|x_j - u_i\|_p.$$

- *$k$-**center**: Same as $k$-median, but we define the cost as*

$$\text{cost}(\mathcal{C}(X)) = \max_{k} \min_{u_i \in \mathbb{R}^d} \max_{j \in C_i} \|x_j - u_i\|_p.$$

- *$k$-**means**: Same as $k$-median, but we instead use the* squared *distances $\|x - u_i\|_p^2$.*

Note that for these clustering problems, the centers $u_i$ do not have to be in $X$ and can be *arbitrary* points in space. That is, once the points are partitioned, we optimize for the choice of centers. Note that even though our initial dataset may satisfy structural properties as sparsity, is it likely that the optimal centers chosen will not. Thus, while our embedding of Theorem C.2 guarantees that $\ell_p$ distances between our dataset is preserved, there is no meaningful notion of what the center is under the embedding. Nevertheless, our embedding implies that the cost of *every* clustering is preserved up to a small multiplicative factor.

**Theorem E.5.** *Consider the $k$-median or $k$-center problem. Let $X$ be a set of $n$ non-negative $s$-sparse vectors in $\mathbb{R}^d$. Let $F : \mathbb{R}^d \to \mathbb{R}^{O(s \log(n)/\varepsilon^2)}$ be the mapping given in Theorem C.2 and $\tilde{X} = \{F(x) \mid x \in X\}$. We have*

$$\Pr\left(\forall \mathcal{C}, \text{cost}(\mathcal{C}(\tilde{X})) = (4 \pm \varepsilon)\text{cost}(\mathcal{C}(X))\right) \geq 1 - 1/\text{poly}(n).$$

*For $k$-means, an identical statement as above holds, except we replace $4$ with $16$.*

*Proof.* Fix a clustering $\mathcal{C}$ (partition of the datapoints). The complication arises due the fact that $\text{cost}(\mathcal{C}(X))$ optimizes for (non-necessarily sparse) centers in $\mathbb{R}^d$ whereas $\mathcal{C}(\tilde{X}))$ optimizes for centers in the projected space. There may not be an analogue of a center chosen in $\mathbb{R}^d$ in the projected space. To get around this, we show we can simply *move* the centers to their closest data point in $X$ with a small multiplicative loss. The proof proceeds formally as follows.

We first only consider the case of $k$-median and $k$-center. Assume that $F$ preserves all pairwise distances between points in $X$, which holds with probability at least $1 - 1/\text{poly}(n)$ and condition on this event. Take any one of the $k$ partitions $C$ of $\mathcal{C}$ and consider the best center $u$ for this partition in $\mathbb{R}^d$. Now let $u'$ be the closest center to $u$ in its partition $C$. Moving $u$ to $u'$ will never decrease the cost (by the optimality of $u$). We claim that it will also never increase the cost by more than a factor of 2. Indeed for $k$-median, the cost increases by a factor of $|C| \cdot \|u - u'\|_p$ by triangle inequality, which is less than $\sum_{j \in C} \|x_j - u\|_p$ by the minimiality of $u'$. Now the sum is just the original cost. For $k$-center a similar reasoning holds.

Call such clusterings where the centers are restricted to the dataset points in $X$ as *basic*. The above reasoning implies that given a partition $\mathcal{C}$, the cost of the optimum clusterings and the optimum basic clusterings only differ by a multiplicative factor of 2. Of course the same reasoning is also true in the projected dimension. But note that the costs of *all* basic clusterings are preserved under $F$, since all pairwise distances are preserved. Thus we have the following chain of inequalities:

$$\text{cost}(\mathcal{C}(\tilde{X}), \text{ basic}) \leq 2\text{cost}(\mathcal{C}(\tilde{X})) \leq 2\text{cost}(\mathcal{C}(\tilde{X}), \text{ basic}) = 2(1 \pm \varepsilon)\text{cost}(\mathcal{C}(X), \text{ basic}),$$

and similarly

$$2(1 \pm \varepsilon)\text{cost}(\mathcal{C}(X), \text{ basic}) \leq 4(1 \pm \varepsilon)\text{cost}(\mathcal{C}(X)) \leq 4(1 \pm \varepsilon)\text{cost}(\mathcal{C}(X), \text{ basic}).$$

Adjusting $\varepsilon$, altogether, this shows that $\text{cost}(\mathcal{C}(\tilde{X}))$ and $\text{cost}(\mathcal{C}(X))$ are within a factor of $4 \pm \varepsilon$ of each other. The argument does not depend on the choice of $\mathcal{C}$ (once we condition on the event that all pairwise distances are preserved). The proof also easily extends to $k$-means, where the basic clusterings now increase the cost by at most a factor of 4 due to the squared triangle inequality.

□

### E.4 DISTANCE ESTIMATION

We first define the distance estimation problem.

**Definition E.4.** *In the distance estimation problem, we want to preprocess a dataset $X$ and output a data structure $\mathcal{D}$. Then on any query $y$, $\mathcal{D}$ outputs an approximation to $\sum_{x \in X} \|x - y\|_p^p$.*

We have the following result.

**Theorem E.6.** *Given a dataset $X \subset \mathbb{R}^d$ of $n$ non-negative $s$-sparse vectors and even integer $p$, we can compute a data structure $\mathcal{D}$ using $O(n \log(n) p s / \varepsilon^2)$ preprocessing time. For any fixed non-negative $s$-sparse query $y$, $\mathcal{D}(y)$ computes a value $t$ satisfying*

$$\left| t - \sum_{x \in X} \|x - y\|_p^p \right| \leq \varepsilon \cdot \sum_{x \in X} \|x - y\|_p^p$$

*with probability $1 - 1/\text{poly}(n)$ with query time $O(\log(n) p s / \varepsilon^2)$.*

*Proof.* We will have $O(\log n)$ independent $d$-variate polynomials $P_1, \ldots P_{O(\log n)}$. Each $P_i$ will output an estimate $t_i$ with the guarantee that

$$\mathbb{E}\left[ \left| t_i - \sum_{x \in X} \|x - y\|_p^p \right| \right] \leq \varepsilon \cdot \sum_{x \in X} \|x - y\|_p^p,$$

which extends to the high-probability guarantee by just taking medians of all the independent estimates.

$P_1$ will be constructed by using the 'base' mapping $f_1 : \mathbb{R}^d \to \mathbb{R}^m$ for $m = O(s/\varepsilon^2)$ of Theorem C.1. We set $\tilde{X}_1 = \{f_1(x) \mid x \in X\}$, and let $P_1(z) = \sum_{x \in \tilde{X}_1} \|x - z\|_p^p$. Then $t_1$ is simply $P_1(f_1(y))$.

Note that each $x \in X$ satisfies $\mathbb{E}[|\|f_1(x) - f_1(y)\|_p^p - \|x - y\|_p^p] \leq \varepsilon \|x - y\|_p^p$ (see the proof of Theorem E.4). Finally, we repeat this independently for all $P_i$. The construction and query times are only dependent on the polynomial evaluation.

Note that each $P_i$ is a polynomial with the number of monomials bounded by $O(mp)$. It takes $O(nmp)$ time to construct $P$ and given $z$, we can compute $P(z)$ in time $O(mp)$ as well. Theorem C.2 guarantees the quality of the approximation.

□

## F USEFUL INEQUALITIES

**Lemma F.1.** *If $X_1, \ldots, X_t$ are i.i.d. Bernoulli$(p)$ for $p \geq 1 - \varepsilon/100$ then*

$$\Pr\left( \left| \sum_{i=1}^t X_i - tp \right| \geq \varepsilon t \right) \leq \exp\left( -\Omega(\varepsilon t) \right).$$

*Proof Sketch.* Consider the complement random variables $Y_i = 1 - X_i$. These are Bernoulli$(1 - p)$ and so by a standard Chernoff bound,

$$\Pr\left( \left| \sum_{i=1}^t Y_i - t(1 - p) \right| \geq \varepsilon t \right) \leq \exp\left( -\Omega(\varepsilon t) \right),$$

as desired. □

**Lemma F.2.** *Let $z \in \mathbb{R}^d$, $p \geq 10 \log(d)/\varepsilon$, and $\varepsilon \in (0, 1/10)$. Then $\|z\|_p \in (1 \pm \varepsilon)\|z\|_\infty$.*

*Proof.* Without loss of generality, suppose $\|z\|_\infty = |z_1| = \max_{i \in [d]} |z_i|$. We have

$$d|z_1|^p \geq \|z\|_p^p \geq |z_1|^p.$$

Taking $1/p$-th powers gives us

$$d^{1/p}|z_1| \geq \|z\|_p \geq |z_1|,$$

and the lemma follows by noting that $d^{1/p} = \exp(\log(d)/p) = \exp(\varepsilon/10) = 1 + \Theta(\varepsilon)$. □

**Lemma F.3.** *If $a, b \in \mathbb{R}^d$ satisfy $\|a - b\|_\infty \leq \delta$, then $|\|a\|_2^2 - \|b\|_2^2| \leq 2\delta\sqrt{d}\|a\|_2 + \delta^2 d$.*

*Proof.* Let $c_i = b_i - a_i$. We know that $|c_i| \leq \delta$. Then $b_i = a_i + c_i$ and $a_i^2 - b_i^2 = a_i^2 - (a_i + c_i)^2 = c_i^2 - 2a_i c_i$ so $|\sum_i (a_i^2 - b_i^2)| \leq d\delta^2 + 2\delta \sum_i |a_i| \leq 2\delta\sqrt{d}\|a\|_2 + \delta^2 d$. □

