# OpenReview forum: "Beyond Worst-Case Dimensionality Reduction for Sparse Vectors"
_ICLR.cc/2025/Conference — ICLR 2025 Poster_

### Official Review · Reviewer_HFgV · 2024-10-31

**Soundness:** 4
**Presentation:** 4
**Contribution:** 4
**Rating:** 8
**Confidence:** 2

**Summary:**

The paper investigates dimensionality reduction techniques for sparse vectors by addressing two central research questions. It explores average-case guarantees, presenting lower bounds that show that essentially the "birthday paradox map" (simply randomly map into the lower dim space with very low chance of coordinate collision) which uses O(s^2) dimension is optimal in many relevant scenarios. In particular, any linear map with similar average-case properties must use Omega(s^2) dimensions; this is then also extended to the case where the map is potentially non-linear but smooth. The authors also propose improved dimensionality reduction for *non-negative* sparse vectors, showing that they can be embedded into fewer dimensions, i.e., O(s log (#vectors) / eps^2) than arbitrary sparse vectors. This shows that non-negativity enables more efficient embeddings, especially in high-dimensional spaces; this is also necessary for smaller embeddings as also shown by the authors. The papers also includes several interesting down-stream applications from optimization.

**Strengths:**

I really like the paper, although I have to admit not being an expert in the field, so I might over-/underestimate the relevance and the first 9 pages only include high-level exposition with no proofs and I did not have the time to check the proofs in details in the appendix. The results seems to be quite "complete" and "complementary" in the sense that basically for each relevant case and bound the corresponding lower bound or construction is presented. The results regarding the additional dimension-reduction in the case of non-negative vectors were somewhat surprising.

It would be great if the paper could give a little bit more intuition why the non-negativity makes such as significant difference. In particular, the authors write that simple translation is not good enough as it might affect sparsity. Intuitively I tend to agree (as this happens to be the case in many similar scenarios).

I think the paper, assuming everything is correct etc, can have quite a few downstream application as dimensionality reduction is at the core of many ML and optimization algorithms.

**Weaknesses:**

Not much to complain about - my only concern or comment is that the first 9 pages only contain high-level summaries. A little bit more detail and intuition, e.g., for the non-negativity induced improvement would have been nice.

**Questions:**

see above

---

> ### Author Response · Authors · 2024-11-20
> **Response to Reviewer HFgV**
>
> Thank you for your feedback.
>
> > It would be great if the paper could give a little bit more intuition why the non-negativity makes such as significant difference. In particular, the authors write that simple translation is not good enough as it might affect sparsity. Intuitively I tend to agree (as this happens to be the case in many similar scenarios).
>
> Intuitively, non-negativity provides more structure that we can exploit for better dimensionality reduction. Consider our map of Theorem 2.5 for the case of the L-infinity norm. Given two vectors x and y, some coordinate witnesses i their L-infinity distance, that is $\|x-y\|_{\infty} = |x_i - y_i|$ We would like to ‘isolate’ this coordinate without knowing the vectors x and y up front (we embed f(x) and f(y) separately). Thus, our map has two important properties:
>
> - (1) Every coordinate in the support of $x$ and $y$ (in particular coordinate $i$) is isolated with large constant probability.
> - (2) If $i$ collides with any other support coordinate, the distance of the embeddings never increases.
>
> By stacking many independent copies of our map, we can boost the success probability of property (1). Furthermore, property (2) ensures that stacking does not distort the distances.
>
> Now property (1) holds for arbitrary sparse vectors, with both positive and negative coordinates. However, for our map, property (2) crucially is only true for non-negative vectors. This is because we take the maximum inside every bucket of coordinates, and $|\max(a_1, \cdots, a_s) - \max(b_1, \cdots, b_s)| \le \max_i |a_i - b_i|$ only holds if the values $a_i$ and $b_i$ are non-negative. For general values, the inequality can only hold up to a factor of 2 which is not strong enough to obtain an embedding. This crisply demonstrates the extra structure we are exploiting.
>
> Now a natural followup question is, are there potentially other maps that would give us dimensionality reduction for arbitrary sparse vectors? Why should we limit ourselves to this particular map in the paper? For that question, we point you to our lower bound of Theorem 2.9. There we show that even for 1-sparse vectors (with positive and negative coordinates), no non-trivial dimensionality reduction can exist in the L-infinity case. In the proof, we use the basis vectors and their negations and crucially exploit cancellations afforded by entries of different signs.
>
> Finally, we note that while a dimensionality reduction map for general sparse vectors (with both positive and negative coordinates) is always valid for non-negative sparse vectors, the converse may not be true. A priori, it seems possible to just use a distance preserving transformation, such as a translation, to convert arbitrary sparse vectors into non-negative sparse vectors. However, a translation which converts arbitrary sparse vectors into non-negative sparse vectors can destroy sparsity.
>
> We refer to the technical overview of Section 3.2 for further details.

---

> > ### Comment · Reviewer_HFgV · 2024-11-27
> >
> > thanks a lot for your answer - i am keeping my score

---

### Official Review · Reviewer_Mwp7 · 2024-11-02

**Soundness:** 3
**Presentation:** 2
**Contribution:** 2
**Rating:** 6
**Confidence:** 3

**Summary:**

The authors focus on the problem of sparse dimensionality reduction in the average case $\ell_p$ norms, where, instead of preserving distances across a high percentage of problem instances, the goal is to preserve distances for a high percentage of pairs of vectors within each instance.

They present a bunch of results for this problem, mainly some are:

1. They show that any linear map, resembling the so called Birthday Paradox map if optimal for the case where $p$ is even. They provide lower bounds to match existing $O(s^2)$ upper bounds on the embedding dimension for $s$-sparse vectors. Their hard cases involves distributions where one first samples co ordinates uniformly at random, and then assigns mean 0 i.i.d. Gaussian value to the non zero co ordinates.
2. They extend this lower bound for weaker guarantees for the case of $\ell_2$ norms, providing stronger results. These extend to continuous, twice differentiable maps, and compositions of such maps that can be interpreted as an encoder-decoder scheme.
3. For the case where the entries could be non negative, they show improved upper bounds by showing the existence of a nonlinear map that achieves a reduced embedding dimension. These results are also paired with lower bounds.

**Strengths:**

Apart from the interesting hard cases, their results on nonlinear mappings for non negative vectors, which provide guarantees for preserving $\ell_\infty$ distances, highlight the importance of moving beyond linear i.i.d. mappings for certain problems.

Their overall theory results are good and I recommend an accept for this venue.

**Weaknesses:**

I don't recommend a strong accept mainly because of lack of a cohesive insight across the two main research questions they address -- average case lower bounds for general sparse vectors and improved upper bounds for non negative sparse vectors.

The paper's array of results could also be supplemented with more insight into the hard distributions (of the type $Unif_{t,r}$) that they constructed. They also mention that Birthday Paradox like maps are folklore, but it would provide more completeness if they referenced mentions or proof of the upper bound. Moreover, their proof techniques might not be highly novel -- for instance the authors ultimately leverage a linear structure of the nonlinear mapping through a Taylor expansion, which is fairly standard.

As a result, while the overall contributions are good, if there were an option for giving a 7, I would give that. However, I do not consider this work sufficiently non trivial to warrant a strong accept.

**Questions:**

Have the authors explored separation results between non-i.i.d. maps and linear i.i.d. maps for this problem?

---

> ### Author Response · Authors · 2024-11-20
> **Response to Reviewer Mwp7, Part 1**
>
> Thank you for your comments. We hope that we can clarify a few points.
>
> > I don't recommend a strong accept mainly because of lack of a cohesive insight across the two main research questions they address -- average case lower bounds for general sparse vectors and improved upper bounds for non negative sparse vectors.
>
>
> Our lower bounds show that for general sparse vectors, even the mild requirement of requiring only a constant fraction of the distances to be preserved already yields strong lower bounds against very general embedding maps. This motivates studying natural/common structure *in addition to sparsity* where dimensionality reduction is possible. A natural structure is non-negativity. Here, we show that we can embed non-negative sparse vectors in very low dimensional space (much lower than what is possible for general sparse vectors), and additionally preserve *all* pairwise distances.
>
> > The paper's array of results could also be supplemented with more insight into the hard distributions (of the type Unif_t,r ) that they constructed.
>
> We would be happy to provide some intuition for our lower bound. The vectors used in the lower bound must represent all possible sparsity patterns. For example, if they were only supported on the first $s$ coordinates, then we could easily obtain a much better upper bound. This motivates us to sample random coordinates to be in our support.
>
> We also put Gaussian entries in the sampled coordinates. This helps us relate $||Ax||_2 \approx ||x||_2$ to an event about Gaussian random variables which can be analyzed. More precisely, in Theorem 3.1 we show that if $ | ||Ax||_2^2 - ||x||_2^2 | \le \gamma ||x\||_2^2$ for a small $\gamma$, then this would imply that a suitable polynomial (obtained by expanding the relation $||Ax||_2^2 - ||x||_2^2$ as a variable in the support of $x$), must lie in a very small interval. This relates to polynomial anti-concentration inequalities for unit variance Gaussians, for which we can invoke Lemma 3.2.
>
> Another useful property of Gaussians that we exploit in our hard distributions is scale invariance:  scaling a vector sampled from Unif_t,r is equivalent to sampling from Unif_t,r′ for an appropriate r′. This is crucial in the proof of Theorem B.2 which extends our average case lower bound to arbitrary smooth functions. Our high level proof idea is to reduce the case of a smooth function f to that of linear maps. This is done by approximating a smooth map via a linear one in a very local region around the origin. The approximation incurs an error, allowing us to essentially only understand the behavior of f(x) for an x also close to the origin, where we can approximate $f(x) \approx Ax$. For this A (derived from f), we know that it approximates the norm of sparse vectors x with very small norm (i.e., those from Unif_t,r for a suitably small r). However by linearity, A must also preserve the norm of any c*x, and by choosing c, it must also preserve the norm of x sampled from Unif_t,1. Now we are in the setting of Theorem 3.1 (linear case) so the lower bound proof goes through.
> They also mention that Birthday Paradox like maps are folklore, but it would provide more completeness if they referenced mentions or proof of the upper bound.
>
> We note that the entire proof of the folklore Birthday paradox map is contained in line 73. We have also updated our paper with citations for prior works that contain the folklore argument.

---

> > ### Author Response · Authors · 2024-11-20
> > **Response to Reviewer Mwp7, Part 2**
> >
> > > Moreover, their proof techniques might not be highly novel -- for instance the authors ultimately leverage a linear structure of the nonlinear mapping through a Taylor expansion, which is fairly standard. As a result, while the overall contributions are good, if there were an option for giving a 7, I would give that. However, I do not consider this work sufficiently non trivial to warrant a strong accept.
> >
> >
> > We would like to highlight that our lower bound proof for average case guarantees contains many non-trivial ideas. First, the lower bound holds for any possible linear map, not just those that do “hashing into buckets.” In other words, the matrix $A$ in our lower bound does not have to have columns of sparsity $1$ (which holds for the birthday paradox). We allow $A$ to have arbitrary real entries and arbitrary density. This leads to complications in our argument since $A$ can spread the mass of a vector along many coordinates. For example, consider the case where $A$ has rows $[1/\sqrt{2}, 1/\sqrt{2}, 0, \cdots,  0]$ and $[1/\sqrt{2}, -1/\sqrt{2}, 0, \cdots , 0]$. These two rows ‘evenly’ spread out the mass on the first two coordinates. That is, $||Ax||_2$ will be exactly equal to $||x||_2$ if $x$ is only supported on the first two coordinates. Thus using arbitrary entries potentially allows the embedding matrix to use multiple coordinates in the embedding vector to `account for’ any coordinate in the input vector $x$. Any lower bound must also rule out these cases.
> >
> > Another complication is that well known methods to prove dimensionality reduction lower bounds do not seem to apply in our setting, because we don't assume all distances have to be preserved. Rather, we assume a weaker hypothesis of only a constant fraction of distances being preserved. Thus, lower bound techniques such as the rank based argument of Alon for the classic JL lemma cannot be directly used. Instead, we generalize rank based arguments to the average case. We show any linear embedding $A$ which preserves most pairwise distances must have a high rank. Unlike the JL case, we only have control over 99% of the entries of A, which is problematic as the remaining 1% can wildly influence the rank to be arbitrarily large (more precisely we look at the rank of $A^TA$). Thus, we consider a “softer” notion of rank, which we show is captured by a suitable polynomial in terms of the entries of $A$ and a sparse vector x which we query $A$ on. We reduce the average case guarantees to showing that the polynomial must lie in a small interval, when we pick x from an appropriate distribution over sparse vectors. Finally, we show that if the rank of $A$ is too small, this polynomial violates anti-concentration properties of random polynomials. We also generalize this argument to sufficiently smooth maps. Please see our technical contribution section, Section 3.2 for more details.
> >
> > In addition, we would like to highlight our upper bound contributions. Linear maps, coming from the JL lemma, are among the most common tools for embeddings. Specifically for sparse vectors, we are not aware of embeddings beyond RIP matrices. We are able to identify an interesting structure, namely non-negatively, which allows us to embed non-negative sparse vectors to a much smaller dimension than what is possible for general sparse vectors. Interestingly, our embedding is non-linear and we show non-linearity is necessary.
> >
> >
> > > Have the authors explored separation results between non-i.i.d. maps and linear i.i.d. maps for this problem?
> >
> > We are not quite sure what you mean by non-i.i.d. and linear i.i.d. maps. Our lower bounds for average case guarantees hold for *any* linear map, not necessarily drawn from a particular distribution. Our upper bounds for non-negative sparse vectors is constructed via a randomized algorithm, but it implies that a map exists, similar to the usual statement of the Johnson Lindenstrauss lemma. It is an interesting future direction to derandomize this construction.
> >
> > Please let us know if you are asking about something else.

---

> ### Author Response · Authors · 2024-11-26
> **Follow up to Reviewer Mwp7**
>
> Thank you again for your review.
>
> We believe we have addressed your concerns by providing insights to connections between our upper and lower bounds as well as a thorough discussion of our lower bounds. If any of your concerns have not been addressed, could you please let us know before the end of the discussion phase?
>
> Many thanks, The authors

---

### Official Review · Reviewer_ASUj · 2024-11-04

**Soundness:** 3
**Presentation:** 3
**Contribution:** 2
**Rating:** 6
**Confidence:** 3

**Summary:**

This paper studies the problem of dimensionality reduction for sparse vectors.

Suppose $x\in \mathbb{R}^d$ is an $s$-sparse vector.
We would like to map $x$ to an $m$-dimensional vector $f(x)$ such that $\lVert f(x) - x \rVert$ is preserved.
Note that this map can be a nonlinear map.
Our goal is to minimize $m$.

This paper provides two main results.
The first one is for the general setting.
The authors show that $m$ needs to be at least $s^2$ for a class of smooth maps $f$.

The second result is for vectors with non-negative entries.
Namely, we further assume that the entries of $x$ are all non-negative.
The authors construct a novel map and show that $s$ is at most $s\log n$ to preserve the pairwise distance among $n$ points.
Also, the authors prove the lower bounds in this setting.

**Strengths:**

The presentation of this paper is generally clear.
Readers from different backgrounds should be able to follow the main idea of this paper.


For the second result, the map constructed by the authors seems to be novel and interesting.
Researchers who work on this area may be able to learn the insight from this construction.

**Weaknesses:**

Regarding the first result, it seems to be natural that one should expect a lower bound of $s^2$ because of the birthday paradox.
Indeed, the main analysis of the lower bound is mostly a probability calculation and the construction is based on uniform sampling which is not particularly surprising.
I am not quite able to see new techniques introduced in this part.

**Questions:**

Line 431: "... a map (randomized) ..." $\to$ "... a (randomized) map ..."?

Line 456: "... (they maybe $0$)." $\to$ "... (they may be $0$)."


Line 842: "... let $i$ be it's ..." $\to$ "... let $i$ be its ..."

Line 908: Should it be the second sum (in the expression for $Z^2$ in line 891)?

Theorem E.6: "datastructure" $\to$ "data structure"

---

> ### Author Response · Authors · 2024-11-20
> **Response to Reviewer ASUj**
>
> Thank you for your comments. We hope that we can clarify a few points.
>
> > Regarding the first result, it seems to be natural that one should expect a lower bound of s^2 because of the birthday paradox. Indeed, the main analysis of the lower bound is mostly a probability calculation and the construction is based on uniform sampling which is not particularly surprising. I am not quite able to see new techniques introduced in this part.
>
> We note that the lower bound holds for any possible linear map, not just those that do “hashing into buckets.” In other words, the matrix $A$ in our lower bound does not have to have columns of sparsity $1$ (which holds for the birthday paradox). We allow $A$ to have arbitrary real entries and $A$ can be arbitrarily dense. This leads to complications in our argument since $A$ can spread the mass of a vector along many coordinates. For example, consider the case where $A$ has rows $[1/\sqrt{2}, 1/\sqrt{2}, 0, …, 0]$ and $[1/\sqrt{2}, -1/\sqrt{2}, 0, …, 0]$. These two rows ‘evenly’ spread out the mass on the first two coordinates. That is, $||Ax||_2$ will be exactly equal to $||x||_2$ if $x$ is only supported on the first two coordinates. Thus using arbitrary entries potentially allows the embedding matrix to use multiple coordinates in the embedding vector to `account for’ any coordinate in the input vector $x$. Any lower bound must also rule out these cases.
>
> Another complication is that well known methods to prove dimensionality reduction lower bounds do not seem to apply in our setting, because we do not assume all distances have to be preserved. Rather, we assume a weaker hypothesis of only a constant fraction of distances being preserved. Thus, lower bound techniques such as the rank based argument of Alon for the classic JL lemma cannot be directly used. Instead, we generalize rank based arguments to the average case. We show any linear embedding A which preserves most pairwise distances must have a high rank. Unlike the JL case, we only have control over 99% of the entries of $A$, which is problematic as the remaining 1% can wildly influence the rank to be arbitrarily large (more precisely we look at the rank of $A^TA$). Thus, we consider a “softer” notion of rank, which we show is captured by a suitable polynomial in terms of the entries of A and a sparse vector $x$ which we query $A$ on. We reduce the average case guarantees to showing that the polynomial must lie in a small interval, when we pick x from an appropriate distribution over sparse vectors. Finally, we show that if the rank of $A$ is too small, this polynomial violates anti-concentration properties of random polynomials. We also generalize this argument to sufficiently smooth maps. Please see our technical contribution section, Section 3.2 for more details.
>
> > Questions and typos.
>
> Thank you for pointing out the small typos. They have been updated.

---

> > ### Author Response · Authors · 2024-11-26
> > **Follow up to Reviewer ASUj**
> >
> > Dear Reviewer ASUj,
> >
> > Thank you again for your review.
> >
> > We have included insights demonstrating some of the technical novelty of our lower bounds. If any of your concerns have not been addressed, could you please let us know before the end of the discussion phase?
> >
> > Many thanks, The authors

---

> > > ### Comment · Reviewer_ASUj · 2024-11-27
> > >
> > > Thanks for the response. I will keep the current rating.

---

### Official Review · Reviewer_e6vD · 2024-11-09

**Soundness:** 3
**Presentation:** 4
**Contribution:** 3
**Rating:** 6
**Confidence:** 4

**Summary:**

The paper studies dimensionality reduction for sparse vectors. Suppose we have a set of s-sparse vectors X in \R^d (i.e., each vector has s non-zero entries), the goal of dimension reduction is to find a transformation f : \R^d -> \R^k, such that k is as small as possible, and || f(x) || \approx ||x|| for "most" x \in X.

Different variants of this basic question are considered:
- if the norm is \ell_p and the goal is to exactly preserve the norms of 99% of the vectors, the authors show a lower bound k = \Omega(s^2). The nice thing here is that it matches the "trivial" birthday paradox based embedding.
- when the norm is \ell_2 and the goal is approximate preservation but to a "small enough" error factor, the authors show that the same lower bound holds.
- the above two lower bounds are for the case when the embedding function f() is assumed to be linear. The authors extend this result to more general non-linear embeddings (assuming smoothness of derivatives), and then argue that a similar lower bound holds on the embedding dimension.

On the upper bound side, the authors consider a set of n vectors, each s-sparse, with _non-negative_ entries. Here, they argue that in order to maintain pairwise distances between these vectors, k = O(s \log n) suffices (with an extra dependence on \epsilon, which is the parameter measuring the approximation error). The statement is interesting, because (a) it overcomes the upper bound shown in the earlier results, and (b) such embeddings are typically difficult for general \ell_p norms (e.g., p=1).

The lower bounds are shown via a nice use of anti-concentration properties of the Gaussian distribution. While not too surprising, it's nice that a simple construction works out. It is nice to see that the lower bound also holds for general smooth embeddings (not just linear). The upper bound is via a non-linear embedding that builds on the birthday construction but with more tricks.

**Strengths:**

- The paper is very well written and the authors do a thorough study of dimension reduction for sparse and non-negative sparse vectors.

- While the proof is not hard, the fact that the lower bound matches the birthday embedding is interesting. It is also interesting that the lower bounds extend to non-linear but smooth embeddings.

- The upper bound also felt interesting, but perhaps the authors can clarify things a bit more (see questions below).

**Weaknesses:**

- The results are nice, but it does feel like a pure theory paper, raising the question of whether ICLR is the right venue. That said, dimension reduction is ubiquitous in ML, and so I feel it is justified.

- The setting considered in the lower bounds and upper bounds are quite different -- lower bounds work only for linear and smooth embeddings while the upper bound uses a non-linear embedding. So stating the results as a "separation" between general and non-negative results does not seem convincing to me. (Of course, there is also the bigger difference that one case considers distances between the vectors, while the other case considers the vectors themselves.)

**Questions:**

- Can the authors say more about whether the known non-linear embeddings (eg., compressed sensing) work for the non-negative vector setting considered in the paper? (will it be that most pairs of distances will be preserved but not all?)

- Please comment on the second point in the weakness above, in case I missed something in the paper.

---

> ### Author Response · Authors · 2024-11-20
> **Response to Reviewer e6vD**
>
> Thank you for your comments. We hope that we can clarify a few points.
>
> > The results are nice, but it does feel like a pure theory paper, raising the question of whether ICLR is the right venue. That said, dimension reduction is ubiquitous in ML, and so I feel it is justified.
>
> Thank you for your feedback. We would like to point out that many papers related to dimensionality reduction have recently appeared in top ML venues. Here is a small selection of papers that study dimensionality reduction and embeddings in a wide range of ML problems within the last 3 years:
>
> - Contrastive dimension reduction: when and how? NeurIPS 24
> - ESPACE: Dimensionality Reduction of Activations for Model Compression. NeurIPS 24
> - Sparse Dimensionality Reduction Revisited. ICML 24
> - Dynamic Metric Embedding into lp Space. ICML 24
> - Simple, Scalable and Effective Clustering via One-Dimensional Projections. NeurIPS 23
> - Fast Optimal Locally Private Mean Estimation via Random Projections. NeurIPS 23
> - SNEkhorn: Dimension Reduction with Symmetric Entropic Affinities. NeurIPS 23
> - A Heat Diffusion Perspective on Geodesic Preserving Dimensionality Reduction. NeurIPS 23
> - Dimensionality Reduction for General KDE Mode Finding. ICML 23
> - A Probabilistic Graph Coupling View of Dimension Reduction. NeurIPS 22
> - Dimensionality reduction for Wasserstein barycenter. NeurIPS 21
> - Randomized dimensionality reduction for facility location and single-linkage clustering. ICML 21
> - HoroPCA: Hyperbolic Dimensionality Reduction via Horospherical Projections. ICML 21
> - Dimensionality Reduction for the Sum-of-Distances Metric. ICML 21
>
>
> > The setting considered in the lower bounds and upper bounds are quite different -- lower bounds work only for linear and smooth embeddings while the upper bound uses a non-linear embedding. So stating the results as a "separation" between general and non-negative results does not seem convincing to me.
>
> Our lower bounds show that for general sparse vectors, even the mild requirement of requiring only a constant fraction of the distances to be preserved already yields strong lower bounds against very general embedding maps. Specifically, this includes the upper bound, which uses a smooth albeit non-linear embedding.
> Thus, this motivates studying structure in addition to sparsity where dimensionality reduction is possible. A natural type of additional structure is non-negativity. Here, we show that we can embed non-negative sparse vectors in very low dimensional space (much lower than what is possible for general sparse vectors), and additionally preserve *all* pairwise distances. We agree that our average case lower bounds are only shown to hold for linear and smooth maps, and it is an interesting future direction to extend our lower bounds to arbitrary maps.
>
> > (Of course, there is also the bigger difference that one case considers distances between the vectors, while the other case considers the vectors themselves.)
>
> Our lower bounds for average case guarantees hold for linear maps and transfer to distances between vectors directly. Here, we are proving a lower bound for linear maps $A$ that have to satisfy $||Ax||_2 \approx ||x||_2$ with probability $99\%$ over the choice of $x$ in a dataset. However, we note that this lower bound also extends to the case where we must satisfy  $||A(x-y)||_2 \approx ||x-y||_2$ for $99\%$ of pairs in a dataset, by just modifying the dataset to contain the difference of all pairwise vectors (due to linearity). In other words, the lower bound we prove is a stronger claim.
>
> > Can the authors say more about whether the known non-linear embeddings (eg., compressed sensing) work for the non-negative vector setting considered in the paper? (will it be that most pairs of distances will be preserved but not all?)
>
> RIP matrices (which are one example of compressed sensing techniques) will give an embedding for all sparse vectors (including nonnegative ones), but it will have an exponential dependence in p on the embedding dimension (See Table 2). Other compressed sensing algorithms such as LASSO are not embeddings.

---

> > ### Author Response · Authors · 2024-11-26
> > **Follow up to Reviewer e6vD**
> >
> > Dear Reviewer e6vD,
> >
> > Thank you again for your review.
> >
> > We believe we have thoroughly addressed your main concerns about the fit of our paper for ICLR and connections between upper and lower bounds. If any of your concerns have not been addressed, could you please let us know before the end of the discussion phase?
> >
> > Many thanks, The authors

---

### Author Response · Authors · 2024-11-20
**Thank you to all reviewers.**

We thank the reviewers for their valuable feedback. Answers are given in a response to each review.

---

### Meta-Review · Area_Chair_JXwF · 2024-12-23

**Metareview:**

The submission analyzes dimension reduction for sparse vectors. It contributes two families of results. The first concerns lower bounds for a type of dimension reduction in which we guarantee to exactly preserve the norms of a 1-\eps fraction of the sparse vectors. For s-sparse vectors, this can be achieved using an embedding in to m >> s^2 dimensions, by randomly mapping coordinates in R^d to coordinates in R^m (the ‘birthday paradox mapping’). The paper proves a lower bound: type of dimension reduction requires s^2 dimensions. These bounds extend to smooth “encoder-decoder” schemes (which contrast with the nonsmooth reconstructions used in compressed sensing). At a technical level, the lower bounds are obtained by choosing the supports uniformly and the nonzero entries from a gaussian distribution, and then applying anti-concentration bounds for the polynomial ||Ax||_2^2 - ||x||_2^2.

The paper also demonstrates improvements for the special case of nonnegative vectors, generating a nonlinear embedding of n vectors into s log (ns) dimensions which preserves all pairwise L^p distances (even for p large).

Reviewers produced a uniformly positive evaluation of the paper noting the following positive points: (i) the paper provides a lower bound confirming an existing intuition for birthday paradox type embeddings of sparse vectors, (ii) the bound is obtained using a nice and relatively clean approach, (iii) the paper’s positive results on nonnegative vectors highlight the importance / role of nonlinear embeddings.

**Additional Comments On Reviewer Discussion:**

The main points raised by the review included the extent to which the lower and upper bounds align (since the lower bound holds for linear or at least sufficiently smooth maps, while the upper bound for nonnegative vectors uses a nonlinear map), and the degree of novelty / significance of the proof technique. Both issues were addressed by the authors response. All four reviewers recommend acceptance.

---

### Decision · Program_Chairs · 2025-01-22

Accept (Poster)